



# Limitations of Emergent Constraints on Multi-Model Projections: Case Study of Constraining Vegetation Productivity With Observed Greening Sensitivity

Alexander J. Winkler[1,2], Ranga B. Myneni[3], and Victor Brovkin[1]

[1]Max Planck Institute for Meteorology, Bundesstrasse 53, 20146 Hamburg, Germany
[2]International Max Planck Research School on Earth System Modelling, Bundesstrasse 53, 20146 Hamburg, Germany
[3]Department of Earth and Environment, Boston University, Boston, Massachusetts 02215, USA

**Correspondence:** Alexander J. Winkler (alexander.winkler@mpimet.mpg.de)

**Abstract.**
Recent research on Emergent Constraints (EC) has delivered promising results. The method utilizes a measurable variable
(predictor) from the recent historical past to obtain a constrained estimate of change in a difficult-to-measure variable (pre-
dictand) at a potential future $CO_2$ concentration (forcing) from multi-model projections. This procedure critically depends
on, first, accurate estimation of the predictor from observations and models, and second, on a robust relationship between
inter-model variations in the predictor-predictand space. We investigate issues related to these two themes in this article, using
vegetation greening sensitivity to $CO_2$ forcing during the satellite era as a predictor of change in Gross Primary Productivity
(GPP) of the Northern High Latitudes region ($60°$ N $- 90°$ N, NHL) for a doubling of pre-industrial $CO_2$ concentration in the
atmosphere. Greening sensitivity is defined as changes in annual maximum of green leaf area index ($LAI_{max}$) per unit $CO_2$
forcing realized through its radiative and fertilization effects. We first address the question of how to realistically characterize
the greening sensitivity of a large area, the NHL, from pixel-level $LAI_{max}$ data. This requires an investigation into uncertain-
ties in $LAI_{max}$ data source and an evaluation of the spatial and temporal variability in greening sensitivity to forcing in both
the data and model simulations. Second, the relationship between greening sensitivity and $\Delta GPP$ across the model ensemble
depends on a strong coupling among simultaneous changes in GPP and $LAI_{max}$. This coupling depends in a complex manner
on the magnitude (level), time-rate of application (scenarios) and effects (radiative and/or fertilization) of $CO_2$ forcing. We
investigate how each one of these three aspects of forcing can impair the EC estimate of the predictand ($\Delta GPP$). Accounting
for uncertainties in greening sensitivity and stability of the relation between inter-model variations results in a quantitative
estimate of the uncertainty ($\pm$ 0.2 Pg C yr$^{-1}$) on constrained GPP enhancement ($\Delta GPP$ = +3.4 Pg C yr$^{-1}$) for a doubling
of pre-industrial atmospheric $CO_2$ concentration in NHL. This $\Delta GPP$ is 60% larger than the conventionally used average of
model projections. The illustrated sources of uncertainty and limitations of the EC method go beyond carbon cycle research
and are generally relevant for Earth system sciences.
*Copyright statement.*



## 1    Introduction

Earth system models (ESMs) are powerful tools to predict response to a variety of forcings such as increasing atmospheric concentration of greenhouse gases and other agents of radiative forcing (Klein and Hall, 2015). Still, longterm ESM projections of climate change can have substantial uncertainties. This can be due to poorly understood processes in some cases, and in others, to missing or simplified representations called parameterizations (Flato et al., 2013; Klein and Hall, 2015; Knutti et al., 2017). Certain important processes, especially in the atmosphere, happen at spatial scales finer than can be possibly represented in current ESMs. Consequently, certain key aspects of the system, such as variability, extreme precipitation events and large-scale climate modes, can be poorly simulated (Flato et al., 2013). Errors propagate and can be amplified through feedbacks among interacting components in the Earth system, resulting in biases whose origins can be difficult to identify (Flato et al., 2013). Furthermore, an inherent component of the Earth climatic system, its internal natural variability, is complicated to represent and simulate in models (Flato et al., 2013; Klein and Hall, 2015).

Model Intercomparison Projects aim is to explore these uncertainties by coordinating a wide range of simulation setups focusing on internal variability, boundary conditions, parameterizations, etc. (Taylor et al., 2012; Eyring et al., 2016; Flato et al., 2013; Knutti et al., 2017). Models developed at various institutions are driven with the same forcing information (e.g. historical forcing) or with identical idealized boundary conditions. However, each modeling group decides which of the processes to consider and implement in their ESM. The conventional approach of handling these multi-model ensembles is to use unweighted ensemble averages (Knutti, 2010; Knutti et al., 2017). This assumes that the models are independent of one another and equally good at simulating the climate system (Flato et al., 2013; Knutti et al., 2017). The large spread between model projections suggests that this assumption is not valid. Therefore, alternate methods have been developed to extract results more accurate than multi-model averages (e.g., model weighting scheme based on preformance and interdependence, Knutti et al., 2017). The concept of *Emergent Constraints* arises in this context, namely, a method to reduce uncertainty in ESM projections relying on historical simulations and observations (Hall and Qu, 2006; Boé et al., 2009; Cox et al., 2013; Klein and Hall, 2015; Cox et al., 2018).

The two key parts of an Emergent Constraint (EC) based method are a linear relationship arising from the collective behavior of a multi-model ensemble and an observational estimate for imposing the said constraint (Fig. 1). The linear relationship is a physically (or physiologically) based correlation between inter-model variations in an observable entity of the contemporary climate system (*predictor*) and a projected variable (*predictand*) that is usually difficult to observe. Combining the emergent linear relationship with observations of the predictor sets a constraint on the predictand (Knutti et al., 2017; Klein and Hall, 2015; Cox et al., 2013; Flato et al., 2013). Many such ECs have been identified and reported, as briefly summarized below.

Hall and Qu (2006) proposed a constraint on projections of snow-albedo feedback based on the correlation between large inter-model variations in feedback strength of the current seasonal cycle. The EC was first established for the CMIP3 ensemble and confirmed for phase five of the Coupled Model Intercomparison Project (CMIP5) (Qu and Hall, 2014; Flato et al., 2013). Several EC studies followed with the goal of reducing uncertainty in projections of the cloud feedback under global warming, as reviewed by Klein and Hall (2015). It is thought that erroneous representation of low-cloud feedback in ESMs contributes



essentially to the large uncertainty in equilibrium climate sensitivity (ECS, 1.5 to 5 K), i.e. warming for a doubling of pre-
industrial atmospheric $CO_2$ concentration ($2\times CO_2$) (Klein and Hall, 2015; Sherwood et al., 2014). Recently, Cox et al. (2018)
presented a different approach to constrain ECS based on its relationship to variability of global temperatures during the recent
historical warming period. They report a constrained ECS estimate of 2.8 K for $2\times CO_2$ (66% confidence limits of 2.2 – 3.4
K).
The concept of EC also found its way into the field of carbon cycle projections. A series of studies analyzed the extent
to which inter-annual atmospheric $CO_2$ variability can serve as a predictor of longterm temperature sensitivity of terrestrial
tropical carbon storage. Cox et al. (2013) and Wenzel et al. (2014) reported an emergent linear relationship, although with
different slopes for CMIP3 and CMIP5 ensembles, resulting in slightly divergent constrained estimates (CMIP3: -53 $\pm$ 17
Pg C $K^{-1}$, CMIP5: -44 $\pm$ 14 Pg C $K^{-1}$). Wang et al. (2014) however were unable to detect a similar relationship between
the proposed predictor and predictand. Recently, Lian et al. (2018) presented an EC estimate of the global ratio of transpiration
to total terrestrial evapotranspiration (T/ET), which is substantially higher (0.62 $\pm$ 0.06) than the unconstrained value (0.41 $\pm$
0.11). For the marine tropical carbon cycle, Kwiatkowski et al. (2017) identified an emergent relationship between the longterm
sensitivity of tropical ocean net primary production (NPP) to rising sea surface temperature (SST) in the equatorial zone and
the interannual sensitivity of NPP to El Niño/Southern Oscillation driven SST anomalies. Tropical NPP is projected to decrease
by 3 $\pm$ 1% for 1 K increase in equatorial SST according to the observational constraint.
Similar results were reported for extra-tropical terrestrial carbon fixation in a $2\times CO_2$ world. Plant productivity is expected
to increase due to the fertilizing and radiative effects of rising atmospheric $CO_2$ concentration. Wenzel et al. (2016) focused
on constraining the $CO_2$ fertilization effect on plant productivity in the northern high latitudes (60° N – 90° N, NHL) and
the entire extra-tropical area in the northern hemisphere (30° N – 90° N) using the seasonal amplitude of longterm $CO_2$
measurements at different latitudes. They presented a linear relationship between the sensitivity of $CO_2$ amplitude to rising
atmospheric $CO_2$ concentration and the relative increase in zonally averaged gross primary production (GPP) for $2\times CO_2$. The
observed $CO_2$ amplitude sensitivities at respective stations provided a constraint on the strength of the $CO_2$ fertilization effect,
namely 37% $\pm$ 9% and 32% $\pm$ 9% for the NHL and the extra-tropical region, respectively.
Focusing on the NHL, Winkler et al. (2018) investigated how both effects of $CO_2$ enhance plant productivity while assess-
ing the feasibility of vegetation greenness changes as a constraint (Fig. 1). Enhanced GPP due to the physiological effect and
ensuing climate warming is indirectly evident in large-scale increase in summer time green leaf area (Myneni et al., 1997; Zhu
et al., 2016). Historical CMIP5 simulations show that the maximum annual leaf area index ($LAI_{max}$, leaf area per ground area)
increases linearly with both $CO_2$ concentration and growing degree days (above 0°C, GDD0) in NHL. To avoid redundancy
from co-linearity between the two driver variables, but retain their underlying time-trend and interannual variability, the dom-
inant mode from a principal component analysis of $CO_2$ and GDD0 was used as the proxy driver (denoted $\omega$). This greening
sensitivity (i.e. $\frac{\Delta LAI_{max}}{\Delta \omega}$) can be inferred for the overlapping historical period from simulations and observations alike. In all
ESMs, changes in GPP arising from the combined radiative and physiological effects of $CO_2$ enrichment strongly correlate
with changes in $LAI_{max}$ in the historical simulations. Thus, the large variation in modelled historical $LAI_{max}$ sensitivities lin-
early maps to variation in $\Delta GPP$ at $2\times CO_2$. Hence, this linear relationship in inter-model variation between $\Delta GPP$ at $2\times CO_2$





and historical greening sensitivities allows using the observed sensitivity as an EC on $\Delta$GPP at $2\times CO_2$ in NHL ($3.4 \pm 0.2$
Pg C yr$^{-1}$, Winkler et al., 2018).
The EC method (Fig. 1) has its limitations. For example, Cox et al. (2013), Wang et al. (2014) and Wenzel et al. (2015)
investigated on constraining future terrestrial tropical carbon storage using the same set of models and data. However, they
arrived at different EC estimates and divergent conclusions. Some reasons for the failure and essential criteria required for
successful application of the EC approach were described previously (Bracegirdle and Stephenson, 2012b; Klein and Hall,
2015), but this list is far from complete. The main focus thus far has been on caveats establishing an emergent linear relationship
in a multi-model ensemble. However, large uncertainty on the constraint could result potentially from how the observational
predictor is derived and compared to the modeled estimates. Here, we revisit the study of Winkler et al. (2018) and elaborate
on key issues concerning sources of uncertainty regarding the constraint and applicability of the EC method.
Uncertainty on the constrained estimate depends on (a) observed predictor and (b) modeled relationship, aside from the
goodness-of-fit of the latter (green shading in Fig. 1). As for (a), the source of observations is an obvious first line of inquiry
(Sect. 3.1). Spatial aggregation of data and model simulations introduces uncertainties, as the EC method is applied on large
areal values of predictor and predictand. This is the subject of Sect. 3.2. The observed and modeled predictors are from the
historical period. The representativeness, duration and match between data and models all introduce an uncertainty related
to variations in the temporal domain – these are explored in (Sect. 3.3). The yellow shading in Fig. 1 represents the total
uncertainty on observed predictor from these three fronts. Regarding (b), the modeled linear relation varies (grey shading in
Fig. 1) depending on three attributes of the forcing, i.e. $CO_2$ concentration change, its magnitude, rate and effect (Sect. 3.4 and
3.5). Lessons learned from analyses along these lines are presented in the conclusion section at the end.



## 2 Data and Methods

### 2.1 Observational data sets

#### 2.1.1 Remotely sensed leaf area index

We used the recently updated version (V1) of the leaf area index data set (LAI3g) developed by (Zhu et al., 2013). It was generated using an artificial neural network (ANN) and the latest version (third generation) of the Global Inventory Modeling and Mapping Studies group (GIMMS) Advanced Very High Resolution Radiometer (AVHRR) normalized difference vegetation index (NDVI) data (NDVI3g). The latter have been corrected for sensor degradation, inter-sensor differences, cloud cover, observational geometry effects due to satellite drift, Rayleigh scattering and stratospheric volcanic aerosols (Pinzon and Tucker, 2014). This data set provides global and year-round LAI observations at 15-day (bi-monthly) temporal resolution and 1/12 degree spatial resolution from July 1981 to December 2016. Currently, this is the only available record of such length.

The quality of previous version (V0) of LAI3g data set was evaluated through direct comparisons with ground measurements of LAI and indirectly with other satellite-data based LAI products, and also through statistical analysis with climatic variables, such as temperature and precipitation variability (Zhu et al., 2013). The LAI3gV0 data set (and related fraction vegetation-absorbed photosynthetically active radiation data set) has been widely used in various studies (Anav et al., 2013; Forkel et al., 2016; Zhu et al., 2016; Mao et al., 2016; Mahowald et al., 2016; Piao et al., 2014; Poulter et al., 2014; Keenan et al., 2016). The new version, LAI3gV1, used in our study is an update of that earlier version.

We also utilized a more reliable but shorter data set from the Moderate Resolution Imaging Spectroradiometer (MODIS) aboard the NASA's Terra satellite (Yan et al., 2016a, b). These data are well calibrated, cloud-screened and corrected for atmospheric effects, especially tropospheric aerosols. The sensor-platform is regularly adjusted to maintain a precise orbit. All algorithms, including the LAI algorithm, are physics-based, well-tested and currently producing sixth generation data sets. The data set provides global and year-round LAI observations at 16-day (bi-monthly) temporal resolution and 0.05° spatial resolution from 2000 to 2016.

Leaf area index is defined as the one-sided green leaf area per unit ground area in broadleaf canopies and as one-half the green needle surface area in needleleaf canopies in both observational and CMIP5 simulation data sets. It is expressed in units of $m^2$ green leaf area per $m^2$ ground area. Leaf area changes can be represented either by changes in annual maximum LAI ($LAI_{max}$) (Cook and Pau, 2013), or growing season average LAI. In this study, we use the former because of its ease and unambiguity, as the latter requires quantifying the start- and end-dates of the growing season, something that is difficult to do accurately in NHL (Park et al., 2016) with the low resolution model data. Further, $LAI_{max}$, is less influenced by cloudiness and noise; accordingly, it is most useful in investigations of long-term greening and browning trends. The drawback of $LAI_{max}$, is the saturation effect at high LAI values (Myneni et al., 2002). However, this is less of a problem in high latitudinal ecosystems which are less-densely vegetated, with $LAI_{max}$, values typically in the range of 2 to 3.

The bi-monthly satellite data sets were merged to a monthly temporal resolution by averaging the two composites in the same month and bi-linearly remapped to the resolution of the applied reanalysis product (0.5°×0.5°, CRU TS4.01).



## 2.1.2 Environmental driver variables

We use temperature, precipitation and $CO_2$ data to derive the observed historical forcing (Sect. 2.3) and to calculate climatic regimes (Fig. 2). Monthly averages of near-surface air temperature and precipitation are from the latest version of the Climatic Research Unit Timeseries data set (CRU TS4.01). The global data are gridded to $0.5° \times 0.5°$ resolution (Harris et al., 2013). Global monthly means of atmospheric $CO_2$ concentration are from the GLOBALVIEW-CO2 product (obspack_co2_1_GLOBALVIEWplus_v2.1_2016_09_02; for details see http://dx.doi.org/10.15138/G3259Z) provided by the National Oceanic and Atmospheric Administration / Earth System Research Laboratory (NOAA / ESRL).

## 2.2 Earth system model simulations

We analyzed recent climate-carbon simulations of seven ESMs participating in the fifth phase of the Coupled Model Intercomparison Project, CMIP (Taylor et al., 2012). The model simulated data were obtained from the Earth System Grid Federation, ESGF (https://esgf-data.dkrz.de/projects/esgf-dkrz/). Seven ESMs provide output for the variables of interest (GPP, $CO_2$, LAI, and near-surface air temperature) for simulations titled esmHistorical, RCP4.5, RCP8.5, 1pctCO2, esmFixClim1, and esmFdbk1. It is the same set of models analyzed in Wenzel et al. (2016) and Winkler et al. (2018).

The esmHistorical simulation spanned the period 1850 to 2005 and was driven by observed conditions such as solar forcing, emissions or concentrations of short-lived species and natural and anthropogenic aerosols or their precursors, land use, anthropogenic as well as volcanic influences on atmospheric composition. The models are forced by prescribed anthropogenic $CO_2$ emissions, rather than atmospheric $CO_2$ concentrations.

Several Representative Concentration Pathways (RCPs) have been formulated describing different trajectories of greenhouse gas emissions, air pollutant production and land use changes for the 21st century. These scenarios have been designed based on projections of human population growth, technological advancement and societal responses (Vuuren et al., 2011; Taylor et al., 2012). We analyzed simulations forced with specified concentrations of a high emissions scenario (RCP8.5) and a medium mitigation scenario (RCP4.5) reaching a radiative forcing level of 8.5 and 4.5 W m$^{-2}$ at the end of the century, respectively. These simulations were initialized with the final state of the historical runs and spanned the period 2006 to 2100.

1pctCO2 is an idealized fully coupled carbon-climate simulation initialized from a steady state of the preindustrial control run and atmospheric $CO_2$ concentration prescribed to increase 1% yr$^{-1}$ until quadrupling of the preindustrial level. The simulations esmFixClim and esmFdbk are set up similar to the 1pctCO2 with the difference, that in esmFixClim (esmFdbk) only the radiative effect from increasing $CO_2$ concentration is included, while the carbon cycle sees the preindustrial $CO_2$ level (*vice versa*) (Taylor et al., 2009, 2012; Arora et al., 2013).

## 2.3 Estimation of greening sensitivities

We largely follow the methodology detailed in Winkler et al. (2018). For both model and observational data, the two-dimensional global fields of LAI and the driver variables are cropped according to different classification schemes (namely, vegetation classes (Olson et al., 2001), climatic regimes, and latitudinal bands). The aggregated values are area-weighted, averaged in



space, and temporally reduced to annual estimates dependent on the variable: annual maximum LAI, annual average atmo-
spheric $CO_2$ concentration, and growing degree days (GDD0, yearly accumulated temperature of days where near-surface air
temperature $> 0°$ C).
We use a standard linear regression model to derive the greening sensitivity. On a global scale, $LAI_{max}$ is assumed to
be a linear function of atmospheric $CO_2$ concentration. For the temperature-limited high northern latitudes, we also have to
account for warming and include temperature as an additional driver. We do this using GDD0. We derive the dominant mode
(denoted $\omega$) through a principal component analysis of $CO_2$ and GDD0 to avoid redundancy from co-linearity between the
two driver variables, but retain their underlying time-trend and interannual variability. Thus, NHL $LAI_{max}$ is formulated as a
linear function of the proxy driver time series $\omega$. The best-fit gradients and associated standard errors of the linear regression
model represent the $LAI_{max}$ sensitivities, or greening sensitivities, and their uncertainty estimates, respectively. For variations
of finer spatial scale, the greening sensitivity is similarly calculated at the pixel scale.



## 3 Results and Discussion

There are two parts to the EC methodology (Fig. 1) – a statistically robust relationship between modeled matching pairs of predictor-predictand values and an observed value of the predictor. The predictors are from a representative historical period. The predictands are modeled changes in a variable of interest at a potential future state of the system, typically one that is difficult to measure. The projection of the observed predictor on the modeled relation yields a constrained value of the predictand. A causal basis has to buttress the predictor-predictand relationship, else the EC method may be spurious. For example, meaningful coupling between concurrent changes in GPP and $LAI_{max}$ with increasing atmospheric $CO_2$ concentration underpins our specific case study, i.e. some of the enhanced GPP due to rising $CO_2$ concentration is invested in additional green leaves by the plants (Winkler et al., 2018). This assures an approximately constant ratio of predictand to predictor across the models within the ensemble, thus setting up the potential for deriving an EC estimate.

Uncertainty on the constrained estimate depends on the observed predictor and modeled relationship, aside from the goodness-of-fit of the latter (Fig. 1). These are detailed below.

### 3.1 Uncertainty in Observed Sensitivity Due to Data Source

We investigate this using LAI data from two different sources, AVHRR (1/12 degree) and MODIS (1/20 degree), and spatially aggregating these by broad vegetation classes, latitudinal bands and climatic regimes. The observed large-area sensitivities are always positive, irrespective of the source data and the method of aggregation (Fig. 2, Tab. 1). This indicates a net increase in green leaf area across the NHL during the observational period, as reported previously (Myneni et al., 1997; Zhu et al., 2016; Forkel et al., 2016). Overall, MODIS based estimates have higher uncertainty because of the shorter length of the data record (17 years). The failure to reliably estimate sensities in tropical forests (also in the latitudinal band 30° S – 30° N, and in hot, wet and humid climatic regimes, see Tab. 1) is due to saturation of optical remote sensing data over dense vegetation ($LAI_{max}$ > 5) and problems associated with high aerosol content and ubiquitous cloudiness. In general, the estimated sensitivities are comparable across sensors and aggregation schemes (e.g. for latitudinal band > 60° N/S, AVHRR: $(3.4 \pm 0.5) \times 10^{-3}$; MODIS: $(3.6 \pm 0.9) \times 10^{-3}$; $LAI_{max}$ ppm$^{-1}$ $CO_2$). However, there are three interesting exceptions. First, higher sensitivities are seen in croplands, which reflect management effects (fertilizer application, irrigation etc.) in addition to $CO_2$ effects (Fig. 2a, Tab. 1). Second, lower sensitivities are seen in sparsely vegetated areas and biomes (low $LAI_{max}$, $\sim$ 1) which are due to nutritionally poor soils and / or inhospitable climatic conditions. Third, similarly low sensitivities are seen in dry regimes where precipitation is limiting and in humid regimes where temperature is limiting (Fig. 2c, Tab. 1).

This analysis illustrates the applicability and limitations of using observed greening sensitivities to $CO_2$ forcing as a constraint on photosynthetic production. For example, data from both AVHRR and MODIS sensors provide a comparable estimate of greening sensitivity in the colder high latitudes (boreal forests and tundra vegetation classes) where precipitation is generally less than 1000 mm (Winkler et al., 2018). However, the remote sensing LAI data are not suitable for similar studies in areas dominated by croplands and in the tropics for reasons stated above.



## 3.2 Uncertainty in Sensitivities Due to Spatial Aggregation

We focus further analyses on the NHL region ($> 60°$ N; Fig. 2b) only because data from both AVHRR and MODIS sensors yield comparable spatially-aggregated greening sensitivities in this region unlike elsewhere, as discussed in Sect. 3.1. In addition to the physiological effect of $CO_2$, also warming plays a key role in controlling plant productivity of these temperature-limited ecosystems, and thus, vegetation greenness. To avoid redundancy from co-linearity between $CO_2$ and GDD0, we reduce dimensionality by performing a principal component analysis of the two driver variables (Sect. 2.3). The resulting first principal component explains most of the variance and retains the trend and year-to-year fluctuations in both $CO_2$ and GDD0. Therefore, we obtain a proxy driver (hereafter denoted $\omega$) that represents the overall forcing signal causing observed vegetation greenness changes in NHL. Accordingly, greening sensitivity for the entire NHL area is derived as response to $\omega$, the combined forcing signal of rising $CO_2$ and warming. This procedure also enables a better comparability between observations and models because varying strengths of physiological and radiative effects of $CO_2$ among models are taken into account (Sect. 3.3 – 3.5).

The vegetated landscape in the NHL region is heterogeneous, with boreal forests in the south, vast tundra grasslands to the north and shrublands in-between. The species within each of these broad vegetation classes respond differently to changes in key environmental factors. Even within a species, such responses might vary due to different boundary conditions, such as topography, soil fertility, micrometeorological conditions, etc. How this fine scale variation in greening sensitivity impacts the aggregated value is assessed below.

The distribution of greening sensitivities from all pixels is slightly skewed towards the positive (blue histogram). The mean value of this distribution (blue dashed line) is comparable to the sensitivity estimate derived from the spatially-averaged NHL time series (yellow dashed line; Fig. 3). Based on the Mann-Kendall test ($p > 0.1$), nearly over half the pixels (54%) show positive statistically significant trends (greening), while about 10% show browning trends (possibly due to disturbances, Goetz et al., 2005). The distribution of these statistically significant sensitivities (red histogram) therefore has two modes, a weak browning and a dominant greening mode, resulting in a substantially higher mean value (red dashed line) in comparison to the spatially-averaged estimate (yellow dashed line; Fig. 3). Thus, by taking into account the remaining 36% of non-significantly changing pixels (as in the NHL spatially-averaged estimate), an additional source of uncertainty is introduced. The mean sensitivity value is, of course, higher when only pixels showing a greening trend are considered in the analysis (green dashed line; Fig. 3). These are the only areas in NHL that actually show a large increase in plant productivity and consequently significant changes in leaf area. ESMs reveal similar pixel-level variation in both $LAI_{max}$ sensitivity and associated changes in GPP in the NHL (Anav et al., 2013, 2015), although ESMs operate on much coarser resolution. Due to the coupling of the predictor and predictand, the distribution of all pixel estimates is approximately the same for the two variables. Accordingly, averaging the equally distributed estimates likely does not affect the predictor-predictand relationship in the model ensemble (Fig. 1). Consequently, if all spatial gridded data arrays are consistently processed to spatially-aggregated estimates, each predictand and predictor (observed and modeled) estimate contain a coherent component of spatial variations. In other words, considering browning and non-significant pixels results in a lower overall $LAI_{max}$ sensitivity in NHL, which in turn leads to a lower constrained estimate of $\Delta$GPP in NHL. This is consistent with the underlying relationship between predictor and predictand. On a



related note, Bracegirdle and Stephenson (2012a) suggest that this source of error is not significantly dependent on the spatial
resolution when comparing model subsets from high to low resolution.
The above analysis informs that spatially-averaged estimates are approximations containing a random error component
due to inclusion of data from insignificantly changing pixels and a systematic bias component from browning pixels. This
uncertainty is relevant to the EC method, where the observed sensitivity decisively determines the constrained estimate from
the ensemble of ESM projections (Winkler et al., 2018; Kwiatkowski et al., 2017). However, if spatial variations are treated
consistently as an inherent component of observations and models, the EC method is only slightly susceptible to this source of
uncertainty.

### 3.3 Uncertainty in Sensitivities Due to Temporal Variations

We seek recourse to longterm CMIP5 ESM simulations covering the historical period 1850 to 2005 (Sect. 2.2) to assess
temporal variation in the predictor variable, because of the shortness of observational record. Three representative models
(CESM1-BGC, MIROC-ESM, and HadGEM2-ES) spanning a broad range of NHL greening sensitivity in the CMIP5 ensemble
(Winkler et al., 2018) are selected for this analysis. For each model, $\text{LAI}_{\text{max}}$ sensitivity to $\omega$ in moving windows of different
lengths (15, 30, and 45 years; Fig. 4) are evaluated. The analysis reveals two crucial aspects that highlight how temporal
variations impair comparability of the predictor variable between models and observations – an essential component of the EC
approach.
First, window locations of modeled and observed predictor variable have to match. If the forcing in the simulations is
low, for example, as in the second half of the 19th century when $CO_2$ concentration was increasing slowly, inter-annual
variability dominates and $\text{LAI}_{\text{max}}$ sensitivity cannot be accurately estimated irrespective of the window length (Fig. 4). With
increasing forcing over time (rising yearly rate of $CO_2$ infusion, and consequently, the concentration), the signal-to-noise
ratio increases and $\text{LAI}_{\text{max}}$ sensitivity to $\omega$ estimation stabilizes, for example, as in the second half of the 20th century.
Therefore, $\text{LAI}_{\text{max}}$ sensitivities estimated at different temporal locations result in non-comparable values and eventually a
false constrained estimate (details in Sect. 3.4). As an example, modeled sensitivities based on a 30-year window centered on
year 1900, when $CO_2$ level increased by 10 ppm, with observed sensitivity estimated from a 30-year window centered on year
2000, when $CO_2$ level increased by 55 ppm, describe different states of the system and therefore should not be used in the EC
method.
Second, in addition to temporal location, window lengths have to match between observations and models. For all three
models, sensitivities estimated from 15-year chunks show high variability and thus, a 15-year record is perhaps too short
to obtain robust estimates. The $\text{LAI}_{\text{max}}$ sensitivity estimation becomes more stable with strengthening forcing and increasing
window length (Fig. 4). As a consequence, using short-term observed sensitivity as a constraint on long-term model projections
results in an incorrect EC estimate. Hence, the MODIS sensor record is, on the one hand, too short and does not, on the other
hand, overlap temporally with the historical CMIP5 forcing (Fig. 1). Therefore, it does not provide a correct observational
constraint.





## 3.4 Level and Time Rate of $CO_2$ Forcing

The EC method raises an obvious question – does it not implicitly assume that the key operative mechanisms underpinning the EC relation remain unchanged because a future system state is being predicted based on its past behavior? To be specific, we are attempting to predict GPP at a future point in time based on greening sensitivity inferred from the past. Does this not require the assumption that the key underlying relationship which makes this prediction possible, namely, a robust coupling between contemporaneous changes in GPP and $LAI_{max}$ remains unchanged from the past to the future? To address this question, we resort to the CMIP5 idealized simulation (1pctCO2), where atmospheric $CO_2$ concentration increases 1% annually, starting from a preindustrial level of 284 ppm until a quadruple of this value is reached (Sect. 2.2). We limit the analysis to the three models (CESM1-BGC, MIROC-ESM, and HadGEM2-ES) which bracket the full range of GPP enhancement and $LAI_{max}$ sensitivity in the original seven ESM ensemble (Winkler et al., 2018).

The relationship between simultaneous changes in GPP and $LAI_{max}$ remains linear for all CMIP5 models in the range $1 \times CO_2$ to $2 \times CO_2$ (Fig. 5, Tab. 2). With concentration increasing beyond $2 \times CO_2$, all models show weakening correlation ($R^2$, Tab. 2) and decreasing slope (*b*, Tab. 2) of this relationship (Fig. 5), suggesting a saturating rate of allocation of additional GPP to new leaves at higher levels of $CO_2$. Consequently, $LAI_{max}$ sensitivity to increasing $CO_2$ and associated warming decreases. At and over $4 \times CO_2$ (1140 ppm), a level unlikely to be seen in the near future, there appears to be no relationship between $\Delta GPP$ and $\Delta LAI_{max}$. This raises the question as to what extent does the weakening of relationship between the predictor and predictand (Fig. 1) at higher $CO_2$ concentration affects the EC analysis. To shed light on this matter, we perform the following *Gedankenexperiment*.

Understanding the relationship and interplay between forcing (increasing $CO_2$ concentration), predictor ($LAI_{max}$ sensitivity), and the predictand ($\Delta GPP$) is key to evaluating the EC method. We conceive four possible scenarios of how the system might behave with increasing forcing. For simplicity, we assume linearly increasing $CO_2$ concentration, use LAI instead of $LAI_{max}$, and GPP refers to its annual value below (Fig. 6). The four scenarios are: *All linear*, *all non-linear* (saturation), and two *mixed linear / non-linear* cases (Tab. A1). We emulate a multi-model ensemble by applying different random parameterizations for the linear and saturation (the hyperbolic tangent function) responses. One of these realizations is assumed to represent pseudo-observations (dashed lines, Fig. 5). We discuss one case in detail for illustrative purposes (No. 3, Tab. A1).

In scenario 3, $\Delta GPP$ increases linearly with increasing $CO_2$ (Fig. 6a), while $\Delta LAI/\Delta GPP$ saturates (Fig. 6b). The LAI sensitivity to $CO_2$ weakens with increasing forcing (Fig. 6c) as a response to saturation of GPP allocation to leaf area. We derive LAI sensitivities to $CO_2$ for three different periods ('past periods' in Fig. 6c) to constrain $\Delta GPP$ at a much higher $CO_2$ level ('projected period' in Fig. 6a). Next, we apply the EC method on these pseudo-projections of $\Delta GPP$ relying on LAI sensitivities derived from the three past periods (Fig. 6d). The EC method is applicable even at a low forcing level (past period 1) in this simplified scenario because we neglect stochastic internal variability of the system. The slope of emergent linear relationship increases (Fig. 6d) as modeled LAI sensitivities decrease with rising $CO_2$ concentration (Fig. 6c). The observational constraint on future $\Delta GPP$, however, remains nearly the same, because pseudo-observed LAI sensitivity also weakens at higher $CO_2$ levels (dashed lines, Fig. 6c, d). Thus, the three EC estimates of $\Delta GPP$ are approximately identical



(Fig. 6d) and independent of the forcing level during past periods. With intensified forcing, the relationship between predictor
and predictand remains linear within the model ensemble, although their relationship becomes non-linear within each model
and, crucially, in reality as well. In other words, as long as the models agree on the occurrence and "timing" of saturation,
changes in predictor and predictand relate linearly within the model ensemble. The same behavior is also seen in the other
three scenarios (Tab. A1; Fig. A1, A2).

6       Nevertheless, with ever increasing forcing and associated steepening of the emergent linear relationship, the LAI sensitivity

loses its explanatory power at some point because the linear relationship eventually lies within the observational uncertainty
and no meaningful constraint can be derived. This and disagreement between models on system dynamics are ultimate limits
of the EC method. Interestingly, we find that all CMIP5 models agree on saturation, but slightly disagree on the timing of
saturation. Further, we find that the 'all non-linear' scenario best describes the dynamics of the system in the forcing range
from $1\times CO_2$ to $4\times CO_2$. However, the saturation of LAI to GPP happens at a lower $CO_2$ level than saturation of GPP to $CO_2$
(Fig. A2). Still, inferences from interpretation of Case 3 (Fig. 6) are equally applicable.
Results from the above *Gedankenexperiment* also highlight the importance of matching window locations and lengths be-
tween models and observations, as discussed earlier (Sect. 3.3). For instance, taking LAI sensitivity from past period 2 (green
dashed line, Fig. 6d) as an observational constraint on the multi-model linear relationship based on past period 3 (red solid line,
Fig. 6d), results in a significant overestimation of constrained $\Delta$GPP (intersection of the two lines, Fig. 6d).
The above analysis informs that the constrained GPP estimate at one future period is nearly independent of the past periods
from when the observational sensitivities are derived, for most realistic scenarios. Now, we evaluate the EC method where
sensitivity from one past period is used to obtain constrained GPP estimates at different periods in the future, i.e. progressively
farther down the time-line. We utilize the greening sensitivity derived from observed $LAI_{max}$ data and apply the EC method to
CMIP5 1pctCO2 simulations. The sensitivities in this case are due to forcing from both $CO_2$ increase and associated warming
during the observational period (Sect. 2.3). We seek constrained GPP estimates at future $CO_2$ levels ($2\times CO_2$, $3\times CO_2$, and
$4\times CO_2$).
Winkler et al. (2018) previously reported a strong linear relationship between modeled contemporaneous changes in $LAI_{max}$
and GPP arising from the combined radiative and physiological effects of $CO_2$ enrichment until $2\times CO_2$ in the CMIP5 ensem-
ble (Fig. 5). As a result, models with low $LAI_{max}$ sensitivity project lower $\Delta$GPP for a given increment of $CO_2$ concentration,
and *vice versa*. Thus, the large variation in modeled historical $LAI_{max}$ sensitivities linearly maps to variation in $\Delta$GPP at
$2\times CO_2$ (Winkler et al., 2018; blue line, Fig. 7a). At higher levels, such as $3\times CO_2$ (green line, $R^2 = 0.93$) and $4\times CO_2$ (red
line, $R^2 = 0.88$), this linear relationship within the model ensemble, while still present, weakens (Fig. 7a; Tab. 3). This is
because the CMIP5 models do not agree on the timing and magnitude of the saturation effect at higher $CO_2$ levels (Fig. 7a).
The increment in constrained GPP estimates for successive equal increments of $CO_2$ decreases due to the saturation effect in
all CMIP5 models (dashed horizontal lines, Fig. 7a). For example, the change in GPP between $3\times CO_2$ and $4\times CO_2$ ($\Delta$GPP
$\sim 1.06$ Pg C yr$^{-1}$, Tab. 3) is much lower than between $2\times CO_2$ and $3\times CO_2$ ($\Delta$GPP $\sim 2.34$ Pg C yr$^{-1}$, Tab. 3).
We have thus far focused on the magnitude of $CO_2$ concentration change and not on the time rate of this change. For
example, a given amount of change in $CO_2$ concentration, say 200 ppm, can be realized over different time periods, say over





a 100 or 150 years. The problem of varying rates of $CO_2$ concentration change is implicitly encountered when ESMs are
executed under different forcing scenarios, such as RCPs. A question then arises whether the constrained GPP estimate is
independent of the time rate of $CO_2$ concentration change and dependent only on the magnitude of $CO_2$ concentration change.
To investigate this aspect of forcing, we extract GPP estimates at the same $CO_2$ concentration (535 ppm; final concentration
in RCP4.5) from three simulations of different forcing rates and calculate the difference relative to a common initial $CO_2$
concentration (380 ppm; initial concentration of RCP scenarios). Hence, the magnitude of the forcing is the same but applied
over different durations (RCP4.5: ∼90yr, RCP8.5: ∼45yr, and 1pctCO2: ∼30yr). A clear majority of the CMIP5 models show
substantial differences in ΔGPP between the different pathways of $CO_2$ forcing. In general, GPP changes are higher for lower
time rates of $CO_2$ forcing, i.e. forcing over longer time periods. As a consequence, the EC estimates of ΔGPP for the same
increase in $CO_2$ concentration are scenario-dependent (Fig. 7b; Tab. 3) – a counter-intuitive result. For instance, ΔGPP in the
low-$CO_2$-rate scenario (RCP4.5: ΔGPP ∼2.84 Pg C yr$^{-1}$, Tab. 3) is ∼39% (1pctCO2: ΔGPP ∼2.05 Pg C yr$^{-1}$, Tab. 3) and
∼20% (RCP8.5: ΔGPP ∼2.38 Pg C yr$^{-1}$, Tab. 3) higher than the high-$CO_2$-rate scenarios for an increase of 155 ppm $CO_2$.
This analysis suggests that the vegetation response to rising $CO_2$ is pathway dependent, at least in the NHL. One of the reasons
for this could be species compositional changes in scenarios of low forcing rates, i.e. over longer time frames. This novel result,
however, requires a separate in-depth study.

## 3.5  Effects of $CO_2$ Forcing

Higher concentration of $CO_2$ in the atmosphere stimulates plant productivity through the fertilization and radiative effects
(Nemani et al., 2003; Leakey et al., 2009; Arora et al., 2011; Goll et al., 2017). The two effects can be disentangled in
the model world by conducting simulations in a '$CO_2$ fertilization effect only' (esmFixClim1) and a 'radiative effect only'
(esmFdbk1) setup (Sect. 2.2). These are termed below as idealized model simulations. We investigate here whether historical
runs and observations, which include both effects, can be used to constrain GPP changes in idealized CMIP5 simulations (e.g.
as in Wenzel et al. (2016)).
We find strong linear relationships between historical $LAI_{max}$ sensitivity and ΔGPP for 2×$CO_2$ in both idealized setups
(esmFixClim1: $R^2$ = 0.92, esmFdbk1: $R^2$ = 0.98, Tab. 3, Fig. 7c). Consequently, this linear relationship is also pronounced for
calculated sums of both effects for each model (esmFixClim1 + esmFdbk1: $R^2$ = 0.95, Tab. 3, Fig. 7c). This suggests that the
two effects act additively on plant productivity and, thus, each effect can be simply expressed in terms of a scaling factor of
the total GPP enhancement. Hence, the application of the EC method on idealized simulations using real world observations is
conceptually feasible.
Interestingly, the two effects contribute about the same to the general increase in GPP at 2×$CO_2$ (esmFixClim1: ΔGPP
∼1.35 Pg C yr$^{-1}$, esmFdbk1: ΔGPP ∼1.38 Pg C yr$^{-1}$, Tab. 3, Fig. 7c). At higher concentrations, such as 3×$CO_2$ and
4×$CO_2$, the enhancement in GPP saturates in both idealized setups. However, the radiative effect becomes dominant relative to
the $CO_2$ fertilization effect when $CO_2$ concentration exceeds 2×$CO_2$ (e.g. at 4×$CO_2$ esmFixClim1: ΔGPP ∼2.42 Pg C yr$^{-1}$,
esmFdbk1: ΔGPP ∼3.06 Pg C yr$^{-1}$, Tab. 3). Therefore, we can expect that at some point in the future, NHL photosynthetic
carbon fixation will benefit more from climate change (e.g. warming) than from the fertilizing effect of $CO_2$.



## 3.6  Uncertainties in the multi-model ensemble

Besides methodological sources of uncertainty discussed above, the estimate of an EC may also be deficient due to inaccurate assumptions about the model ensemble. First, possible common systematic errors in a multi-model ensemble (i.e. the entire ensemble misses an unknown but for the future essential process) are implicitly omitted in the EC approach, however, could cause a general over- or underestimation of the constrained value (Bracegirdle and Stephenson, 2012b; Stephenson et al., 2012). Second, the set of forcing variables for historical simulations may be incomplete (i.e. not yet identified drivers of observed changes) and, thus, the comparability of observations and model simulations is limited (Flato et al., 2013). Third, the EC method can be overly sensitive to individual models of the ensemble, which has a bearing on the robustness of the constrained value (Bracegirdle and Stephenson, 2012b). Bracegirdle and Stephenson (2012b) proposed a diagnostic metric (Cook's distance) to test an ensemble for influential models. Fourth, the assumption behind the predictand-predictor relationship has to rely on a logical connection within the model ensemble, meaning that the analyzed characteristic of the predictor variable (e.g. sensitivity to the forcing, or historical relative/absolute changes) is causally linked to changes in the predictand variable. For instance, Wenzel et al. (2016) reported a linear relationship between relative changes in GPP for doubling of $CO_2$, so changes relative to the preindustrial state, and historical sensitivity of $CO_2$ amplitude to rising $CO_2$, so neglecting the initial state. This statistical relationship can be spurious, because the model skill of simulating an accurate initial state and a plausible sensitivity to a forcing are not connected.

These issues are to be contemplated when establishing an EC estimate and evaluating its robustness.





# 4   Conclusions

An in-depth analysis of the EC method is illustrated in this article through its application to projections of change in NHL photosynthesis under conditions of rising atmospheric $CO_2$ concentration. Key conclusions highlighting the functionality of the EC method are presented below.

The importance of how the observational predictor is obtained cannot be emphasized enough because it essentially defines the constrained estimate. Thus, considerable care is required when selecting and processing the observational datasets. The LAI data products of both AVHRR and MODIS sensors provide comparable estimates of greening sensitivity in the colder northern high latitudes (i.e. boreal forests and tundra vegetation classes). In these ecosystems, factors associated with GPP enhancement from $CO_2$ forcing and consequent investment in leaf area dominate. This is not the case in croplands and tropical areas. Therefore, the use of greening sensitivity as an observational constraint is not feasible in regions where croplands and/or tropical vegetation dominate.

Spatially aggregating observations and model output of different resolutions in the EC method is another source of uncertainty. Regional estimates of greening sensitivity are approximations of complex fine-scale processes. Aggregation will inevitably introduce a random error component due to inclusion of data from areas where LAI is not changing and a systematic bias from areas where LAI is decreasing (browning). The spatially-aggregated greening sensitivity is meaningful only if most of the region is greening in response to $CO_2$ forcing. However, as long as spatial variations in observations and models simulations are treated consistently, this source of uncertainty is likely of minor importance.

A large source of uncertainty is associated with temporal variability of the predictor variable throughout the historical period. The evaluation of greening sensitivity requires temporal window lengths of sufficient duration, approximately 30 years, and location along the forcing time line. And, these should match between models and observations. For example, the analysis in Wenzel et al. (2016) might have yielded different results and conclusions if model and observational predictor sensitivities were temporally matched. The relevance of window length decreases with increasing and accelerating forcing, depending on the magnitude of natural/internal variability (signal-to-noise ratio) of the predictor variable.

The level, effect and duration of $CO_2$ forcing have a bearing on the linear relationship between GPP enhancement and predictor sensitivities (Fig. 1). For example, the relationship underpinning the EC method, namely, that between concurrent $\Delta GPP$ and $\Delta LAI_{max}$, changes with increasing forcing level ($CO_2$ concentration). This relation breaks down at very high $CO_2$ concentrations at which point the EC method fails. The two dominant effects of rising $CO_2$ concentration on vegetation, namely, the fertilization and radiative effects, appear to be approximately additive in terms of GPP enhancement to $CO_2$ forcing. Therefore, the EC method can be applied to constrain estimates of GPP due to one or the other, or both the effects. The models, however, document a higher radiative effect than fertilization at high $CO_2$ concentrations, i.e. $3 \times CO_2$ and higher. An intriguing conclusion from our analysis is that the time-rate of forcing has an effect on GPP changes, that is, the projected GPP enhancement to $CO_2$ forcing seems to be dependent on how the forcing is applied over time, as in different scenarios or RCPs. This aspect is presently not well understood and requires further study.





The analyses and inferences presented in this article lead to the following concrete result. The uncertainty on EC estimate of
GPP enhancement in NHL ($\Delta$GPP = +3.4 Pg C yr$^{-1}$) for a doubling of pre-industrial atmospheric $CO_2$ concentration is $\pm$ 0.2
Pg C yr$^{-1}$ (Winkler et al., 2018). This EC estimate is 60% larger than the conventionally used average of model projections
(44% higher at the global scale), leading Winkler et al. (2018) to conclude that most CMIP5 models included in their analysis
were largely underestimating photosynthetic production.
In this article, we scrutinized potential sources of uncertainty and limitations of the applicability of the EC method. Our
findings are illustrated by means of a case study in carbon cycle research, however, are generally relevant and applicable in
Earth system sciences.



*Author contributions.*   All authors contributed ideas and to writing of the manuscript. A.J.W. did the analysis.
*Competing interests.*   The authors declare that they have no conflict of interest.
*Acknowledgements.*   We thankfully acknowledge T. Park and C. Chen for their help with remote sensing data. We thank G. Lasslop for
reviewing the manuscript. R.B.M. thanks Alexander von Humboldt Foundation and NASA's Earth Science Division for funding support that
made his participation possible in this research.





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





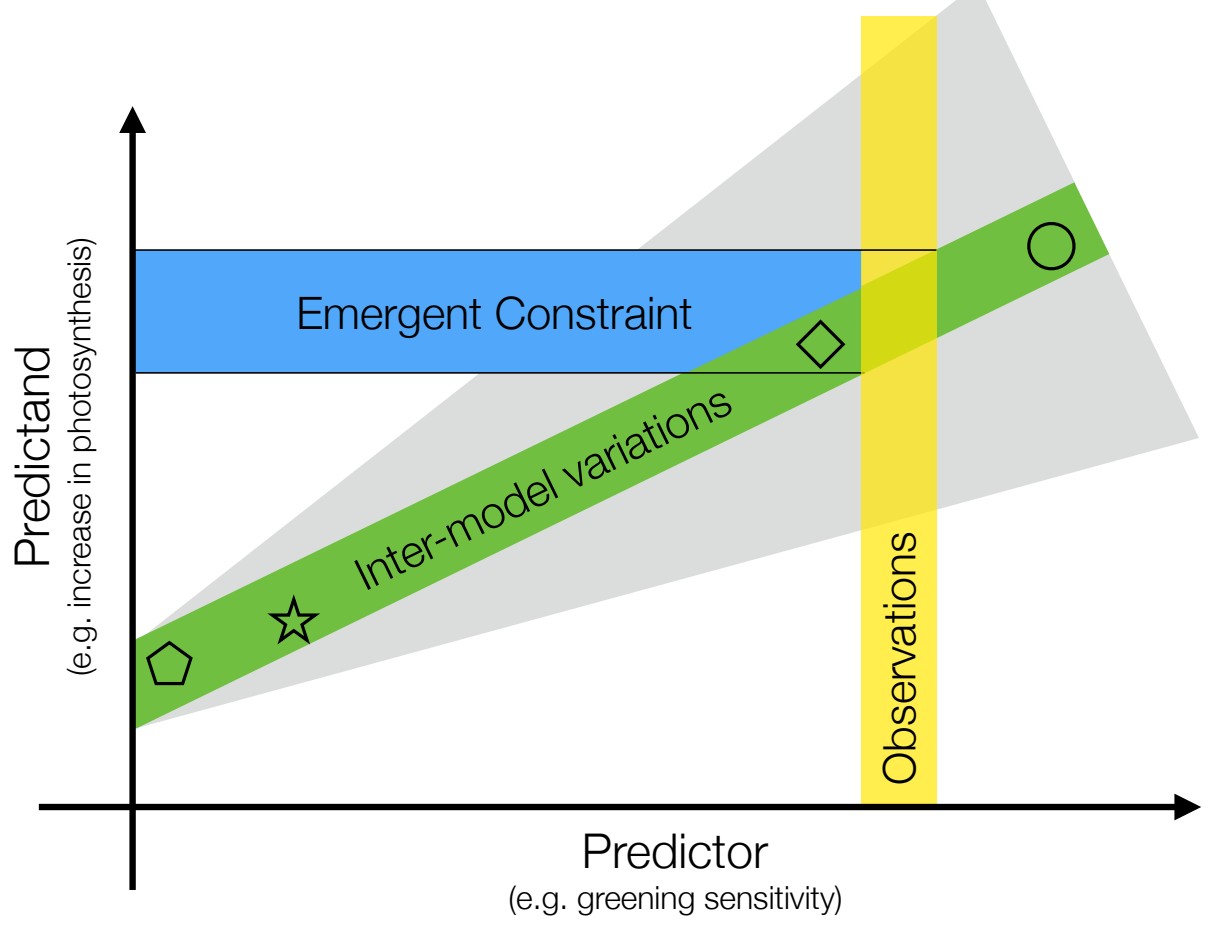

**Figure 1.** Schematic depiction of the Emergent Constraint (EC) method and factors affecting the uncertainty of the constrained estimate.
The predictor (x axis) is change in annual maximum of green leaf area index ($LAI_{max}$) due to unit forcing ($CO_2$ increase and associated
climatic changes) during a representative historical period. It is termed greening sensitivity in this study. The predictand (y axis) is projected
changes in Gross Primary Productivity (GPP) in response to rising $CO_2$ concentration (e.g. for a doubling of the pre-industrial level). Both
the predictor and predictand refer to large area values, in this case, the entire Norther High Latitudes (NHL). Inter-model variations (each
symbol represents a model) in matching pairs of predictor and predictand result in a linear relationship between the two (green band), i.e. the
ratio (predictand/predictor) is approximately constant across the model ensemble. The slope depends on forcing attributes (gray shading),
such as its level ($CO_2$ concentration, Sect. 3.4), time rate of application (scenarios such as various RCPs, Sect. 3.4) and different effects
(i.e. fertilization, radiative, etc., Sect. 3.5). The observed sensitivity (yellow vertical bar) is used to find the constrained estimate of the
predictand (i.e. change in GPP). The ability to accurately estimate the predictor depends on the source of observational data (Sect. 3.1), and
its spatial (Sect. 3.2) and temporal variability (Sect. 3.3). Observed (yellow bar) and modeled predictor values (x coordinate of symbols) must
be obtained from matching time periods, i.e. at the same level of historical forcing, to ensure comparability (Sect. 3.4). All these factors,
together with the goodness-of-fit of inter-model variations (width of green shading), finally define the uncertainty of the derived constrained
estimate (blue horizontal bar with black solid lines depicting the upper and lower bound of uncertainty).




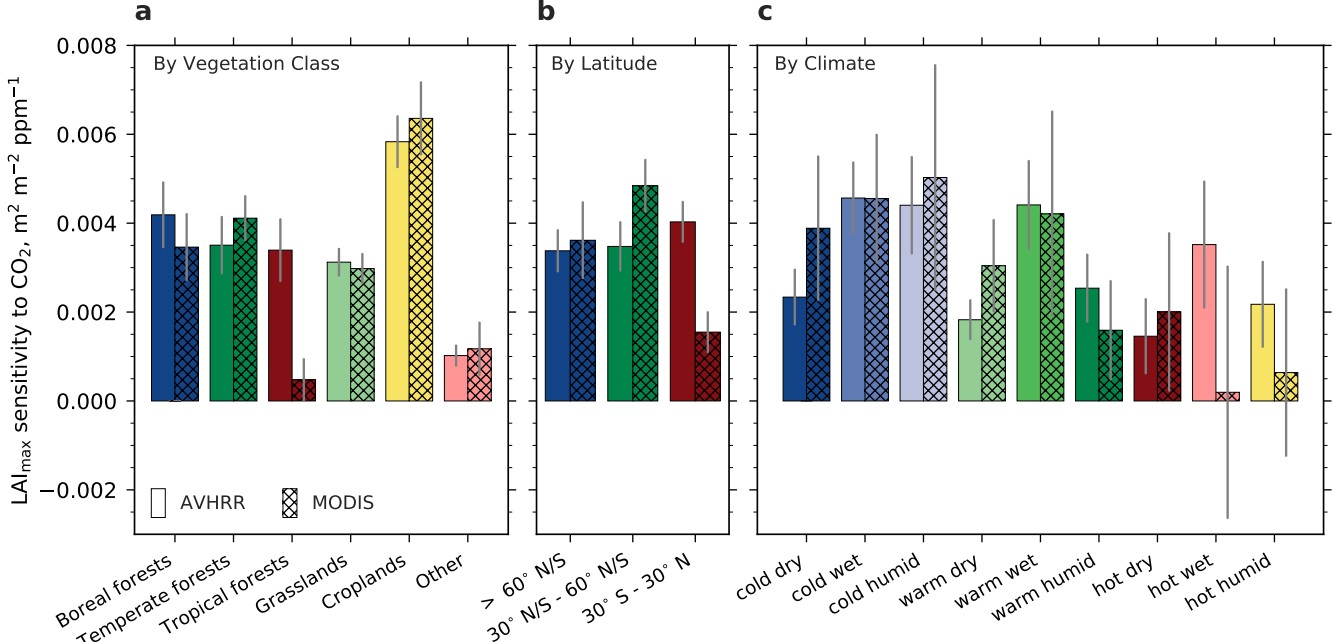

**Figure 2.** Bar charts showing regression slopes of $LAI_{max}$ against atmospheric $CO_2$ concentration for broad vegetation classes (**a**, Olson et al. (2001), latitudinal bands (**b**) and climate regimes (**c**). The class "Other" includes deserts, mangroves, barren and urban land, snow and ice, and permanent wetlands. The climatic boundaries are defined as follows - cold: $< 10°$ C; warm: $> 10°$ C & $< 25°$C; hot: $> 25°$ C; dry: $< 500$ mm a$^{-1}$; wet: $> 500$ mm a$^{-1}$ & $< 1000$ mm a$^{-1}$; humid: $> 1000$ mm a$^{-1}$. Sensitivities evaluated from data from two satellite-borne sensors are shown, AVHRR (1982 – 2016, Pinzon and Tucker (2014)) and MODIS (2000 – 2016, Yan et al. (2016a, b)). Grey bars indicate the standard error of the best linear fit.





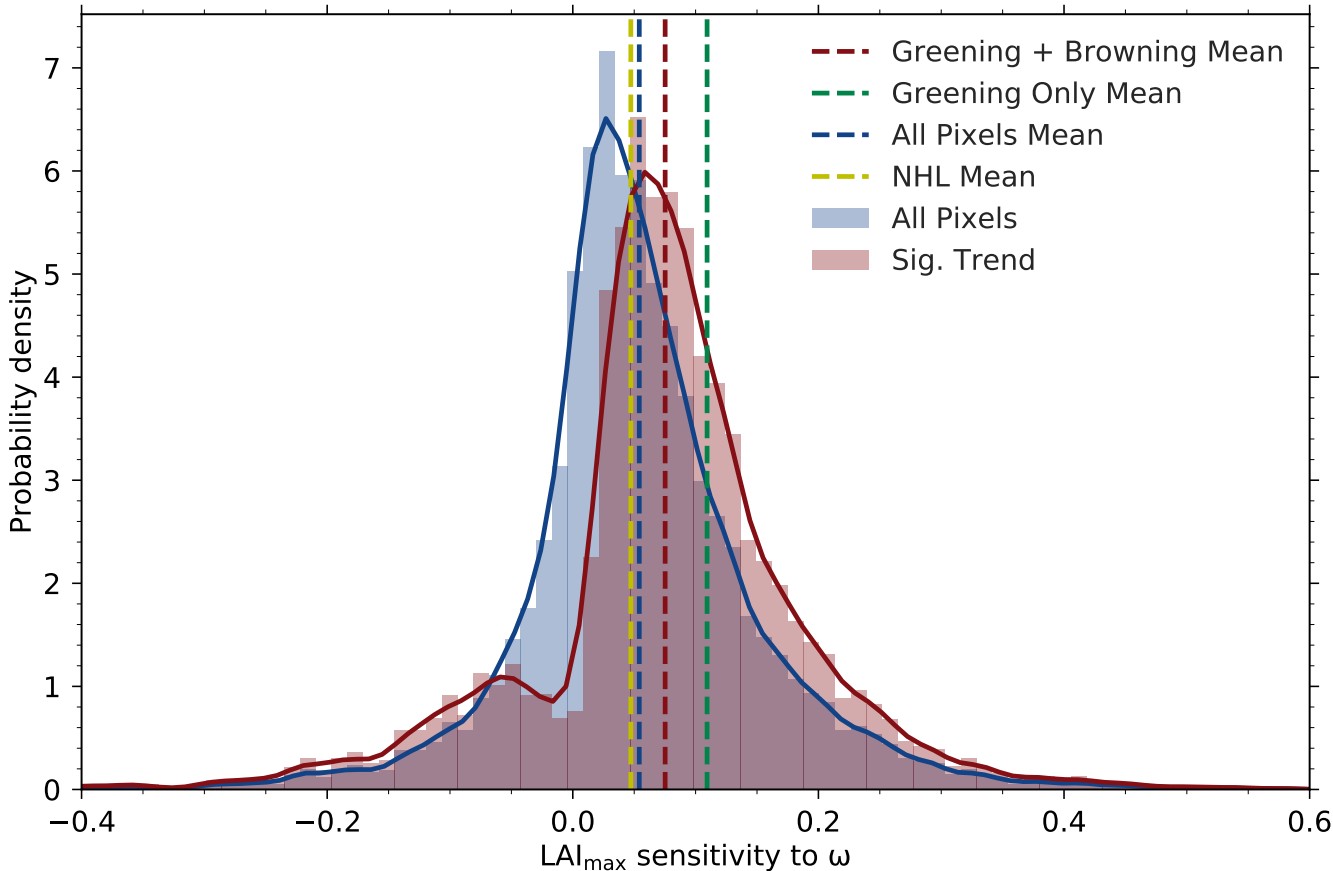

**Figure 3.** Histograms and associated probability density functions (Gaussian kernel density estimation) of observed $LAI_{max}$ sensitivity to $\omega$ at pixel scale for the northern high latitudinal band ($> 60°$ N, data from AVHRR sensor). Blue color depicts the distribution of $LAI_{max}$ sensitivities of all pixels and the red color for pixels with statistically significant (Mann-Kendall test, p < 0.1) greening or browning trends (the dashed lines denote the respective mean value). The green dashed line shows the mean value of 'greening' pixels only, whereas the yellow dashed line shows the $LAI_{max}$ sensitivity to $\omega$ for the entire northern high latitudinal belt.





**Figure 4.** Temporal variation of $LAI_{max}$ sensitivity to $\omega$ in three selected CMIP5 models spanning the full range from low (CESM1-BGC, **a**), to closest-to-observations (MIROC-ESM, **b**), to high-end (HadGEM2-ES, **c**). The colored lines show $LAI_{max}$ sensitivity variations for moving windows of varying length of 15 (blue), 30 (green), and 45 (red) years over the historical period from 1860 to 2005.





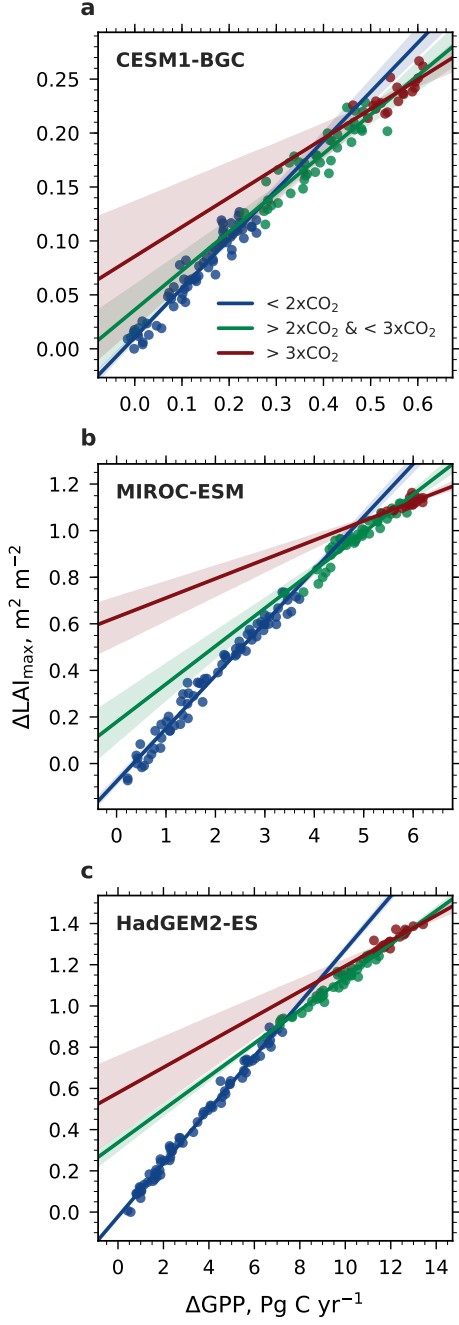

**Figure 5.** Correlation of $\Delta LAI_{max}$ and $\Delta GPP$ with increasing $CO_2$ forcing, starting from a pre-industrial concentration of 280 ppm ($1xCO_2$)

to $4xCO_2$ (CMIP5 1pctCO2 simulations). Results are shown for three selected CMIP5 models spanning the full range of $LAI_{max}$ sensitivity

to $\omega$, low-end: CESM1-BGC (**a**), closest-to-observations: MIROC-ESM (**b**), and high-end: HadGEM2-ES (**c**). Blue colored dots show the

relation between $1xCO_2$ and $2xCO_2$, green colored dots between $2xCO_2$ and $3xCO_2$, and red colored dots between $3xCO_2$ and $4xCO_2$.

The respective colored lines represent the best linear fit through those dots and the shading represents the 95% confidence interval.





**Figure 6.** Gedankenexperiment to examine the applicability of Emergent Constraints analysis under the assumption of an idealized linear / nonlinear behavior of the system (Case 3, Table A1). **a**, Changes in GPP relate linearly to changes in $CO_2$ concentration. The yellow band marks the projection period of interest, i.e. the period of $CO_2$ concentration from $x + 4\Delta$ to $x + 5\Delta$. **b**, The increment in LAI with increasing GPP is assumed to decrease with rising $CO_2$ concentration (described by a hyperbolic tangent function). The parameterization in the linear and nonlinear functions for pseudo observations (dashed black line) as well as models (solid grey lines) are determined randomly for each model. **c**, The diagnostic variable, LAI sensitivity to $CO_2$, is decreasing with increasing $CO_2$ as a consequence of the nonlinear relation between $\Delta$GPP and $\Delta$LAI. The colored bands indicate three 'past' periods from $x$ to $x + \Delta$ (blue), $x + \Delta$ to $x + 2\Delta$ (green), and $x + 2\Delta$ to $x + 3\Delta$ (red). **d**, Linear relationships among the pseudo model ensembles (Ensemble LR, colored lines) between LAI sensitivities to $CO_2$ of the three past periods and $\Delta$GPP from the projected period. Colored dots mark different models and the dashed lines represent associated pseudo observations for the respective historical period. Yellow solid line depicts the constant Emergent Constraint on projected $\Delta$GPP irrespective of the past period.





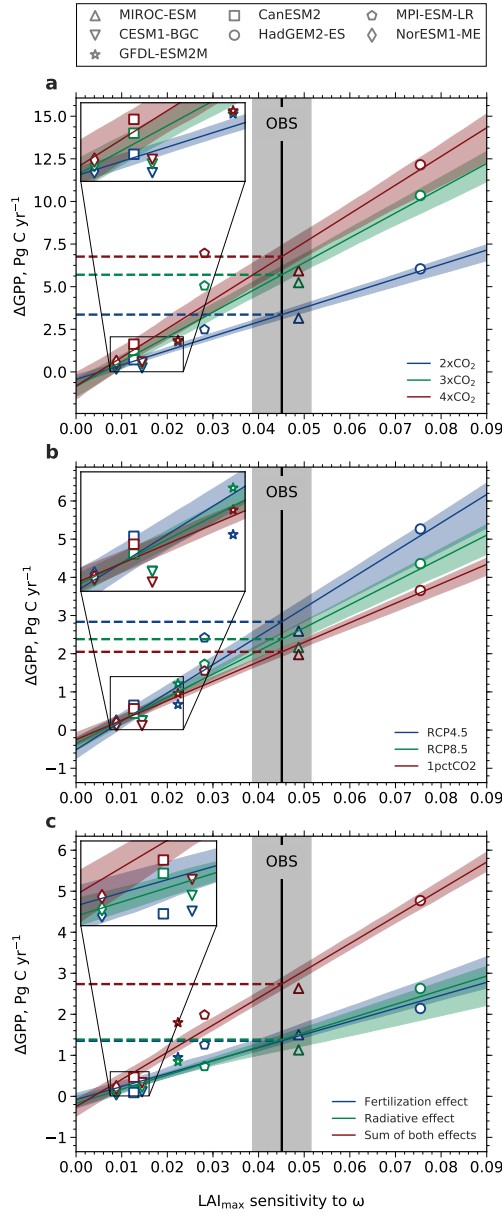

**Figure 7.** Linear relationships between historical sensitivity of $LAI_{max}$ to $\omega$ and absolute increase of GPP at different levels (**a**), different

time-rates (**b**) as well as effects of rising $CO_2$ (**c**). The black solid line depicts the observational sensitivity including the standard error (grey

shading). Each CMIP5 model is represented by a distinct marker (legend at the top). The colored lines show the best linear fits including the

68% confidence interval estimated by bootstrapping across the model ensemble. The colored dashed lines indicate the derived constraints on

$\Delta$GPP. **a**, Absolute changes in GPP at different levels of $CO_2$: $2\times CO_2$ (blue), $3\times CO_2$ (green), and $4\times CO_2$ (red). **b**, Absolute changes in

GPP for rising $CO_2$ concentration from 380 to 535 ppm at different time-rates: RCP4.5 (90 yr, blue), RCP8.5 (45 yr, green), and 1pctCO2

(30 yr, red). **c**, Absolute changes in GPP due to the two disentangled effects of $CO_2$ at $2\times CO_2$ in idealized simulations: Fertilization effect

(esmFixClim1, blue), radiative effect (esmFdbk1, green), and the sum of both effects (red).



1  **Table 1.** Coefficients of determination ($R^2$) of $LAI_{max}$ sensitivity to $CO_2$ for different large-scale aggregated regions. Data are from two

2  optical remote sensors of different time length, AVHRR (1982 – 2016) and MODIS (2000 – 2016). Asterisks denote non-significant values:

3  ** $p > 0.1$; * $p > 0.05$.

| Correlation coefficient $R^2$ | AVHRR | MODIS |
|---|---|---|
| **Biomes** | | |
| Boreal forests | 0.49 | 0.58 |
| Temperate forests | 0.47 | 0.81 |
| Tropical forests | 0.41 | 0.06** |
| Graslands | 0.75 | 0.83 |
| Croplands | 0.75 | 0.8 |
| Other | 0.35 | 0.2* |
| **Latitudinal Bands** | | |
| $> 60°$ N/S | 0.51 | 0.61 |
| $30°$ N/S – $60°$ N/S | 0.67 | 0.83 |
| $30°$ S – $30°$ N | 0.65 | 0.26 |
| **Climate Space** | | |
| cold dry | 0.29 | 0.27 |
| cold wet | 0.49 | 0.4 |
| cold humid | 0.33 | 0.21* |
| warm dry | 0.33 | 0.36 |
| warm wet | 0.37 | 0.18* |
| warm humid | 0.25 | 0.12** |
| hot dry | 0.08* | 0.08** |
| hot wet | 0.15 | 0.00** |
| hot humid | 0.13 | 0.01** |



**Table 2.** Slopes ($b$) and coefficients of determination ($R^2$) for regression between changes of $LAI_{max}$ against changes in annual mean GPP at different atmospheric $CO_2$ levels in all available CMIP5 models (1pctCO2 simulation). Asterisks denote non-significant values: ** $p >$ 0.1; * $p > 0.05$.

| Correlation details | < 2x$CO_2$ | | > 2x$CO_2$ & < 3x$CO_2$ | | > 3x$CO_2$ | |
|---|---|---|---|---|---|---|
| | $b$ | $R^2$ | $b$ | $R^2$ | $b$ | $R^2$ |
| MIROC-ESM | 0.23 | 0.97 | 0.16 | 0.89 | 0.08 | 0.63 |
| CESM1-BGC | 0.45 | 0.93 | 0.36 | 0.82 | 0.27 | 0.62 |
| GFDL-ESM2M | 0.37 | 0.89 | 0.04 | 0.07** | 0.01 | 0.12** |
| CanESM2 | 0.22 | 0.95 | 0.19 | 0.83 | 0.17 | 0.67 |
| HadGEM2-ES | 0.13 | 0.99 | 0.08 | 0.96 | 0.06 | 0.78 |
| MPI-ESM-LR | 0.13 | 0.94 | 0.09 | 0.78 | 0.04 | 0.51 |
| NorESM1-ME | 0.26 | 0.94 | 0.2 | 0.77 | 0.09 | 0.27 |





**Table 3.** Coefficients of determination ($R^2$) of the emergent linear relationships in Figure 7 (asterisks denote non-significant values: ** p >
0.1; * p > 0.05). Emergent Constraints on $\Delta$GPP (upper and lower bound of uncertainty in square brackets) for different atmospheric $CO_2$
levels and fully-coupled as well as idealized setups. The rightmost column shows the increase of $\Delta$GPP for an increment of $1 \times CO_2$. The
lowermost section compares EC estimates of $\Delta$GPP for equivalent changes in $CO_2$ concentration ($CO_2$ rises from 380 to 535 ppm), but for
different time-rates.

| | $R^2$ | EC $\Delta$GPP estimate (Pg C yr$^{-1}$) | EC $\Delta$GPP for $\Delta 1 \times CO_2$ (Pg C yr$^{-1}$) |
|---|---|---|---|
| **2x$CO_2$** | | | |
| Fully coupled (1pctCO2) | 0.96 | 3.36 [3.15, 3.56] | – |
| $CO_2$ fertilization only (esmFixClim1) | 0.88 | 1.35 [1.29, 1.62] | – |
| Radiative effect only (esmFdbk1) | 0.94 | 1.38 [1.13, 1.51] | – |
| Sum of both effects (esmFixClim1 + esmFdbk1) | 0.95 | 2.74 [2.6, 2.9] | – |
| **3x$CO_2$** | | | |
| Fully coupled (1pctCO2) | 0.93 | 5.7 [5.26, 6.16] | 2.34 |
| $CO_2$ fertilization only (esmFixClim1) | 0.92 | 2.15 [2.02, 2.37] | 0.79 |
| Radiative effect only (esmFdbk1) | 0.98 | 2.53 [2.3, 2.66] | 1.15 |
| Sum of both effects (esmFixClim1 + esmFdbk1) | 0.96 | 4.68 [4.38, 4.97] | 1.94 |
| **4x$CO_2$** | | | |
| Fully coupled (1pctCO2) | 0.88 | 6.76 [6.08, 7.53] | 1.06 |
| $CO_2$ fertilization only (esmFixClim1) | 0.88 | 2.42 [2.23, 2.74] | 0.28 |
| Radiative effect only (esmFdbk1) | 0.97 | 3.06 [2.83, 3.2] | 0.53 |
| Sum of both effects (esmFixClim1 + esmFdbk1) | 0.95 | 5.49 [5.09, 5.85] | 0.81 |
| **380 – 535 ppm $CO_2$** | | | |
| Slow increase in $CO_2$ (RCP4.5) | 0.93 | 2.84 [2.54, 3.08] | - |
| Medium-fast increase in $CO_2$ (RCP8.5) | 0.96 | 2.38 [2.18, 2.55] | - |
| Rapid increase in $CO_2$ (1pctCO2) | 0.96 | 2.05 [1.94, 2.16] | - |



**Figure A1.** Gedankenexperiment to examine the applicability of the Emergent Constraints analysis assuming an idealized linear / linear behavior of the system (Case 1, Table A1). **a**, Changes in GPP relate linearly to changes in $CO_2$ concentration. The yellow band marks the projection period of interest, i.e. the period of $CO_2$ concentration from $x + 4\Delta$ to $x + 5\Delta$. **b**, Changes in LAI relate linearly to changes in GPP. The parameterization in the linear functions for pseudo observations (dashed black line) as well as models (solid grey lines) are determined randomly for each model. **c**, The diagnostic variable, LAI sensitivity to $CO_2$, remains constant with increasing $CO_2$ as a consequence of the overall linear characteristics of the system. The colored bands indicate three 'past' periods from $x$ to $x + \Delta$ (blue), $x + \Delta$ to $x + 2\Delta$ (green), and $x + 2\Delta$ to $x + 3\Delta$ (red). **d**, Linear relationships among the pseudo model ensembles (Ensemble LR 1-3 on top of each other, red) between LAI sensitivity to $CO_2$ of the three past periods and $\Delta$GPP from the projected period. Red dots mark different models and the dashed line represents associated pseudo observations for all three historical periods. Yellow solid line depicts the constant Emergent Constraint on projected $\Delta$GPP irrespective of the past period .



**Figure A2.** Gedankenexperiment to examine the applicability of the Emergent Constraints analysis assuming an idealized nonlinear / non-linear behavior of the system (Case 4, Table A1). **a**, $\Delta$GPP decreases with increasing $CO_2$ concentration (described by a hyperbolic tangent function). The yellow band marks the projected period of interest, i.e. the period of $CO_2$ concentration from $x + 4\Delta$ to $x + 5\Delta$. **b**, Also $\Delta$LAI decreases with increasing GPP (described by a hyperbolic tangent function). The parameterization in the hyperbolic tangent functions for pseudo observations (dashed black line) as well as models (solid grey lines) are determined randomly for each model. **c**, The diagnostic variable, LAI sensitivity to $CO_2$, is decreasing with increasing $CO_2$ as a consequence of the overall saturating characteristics of the system. The colored bands indicate three 'past' periods from $x$ to $x + \Delta$ (blue), $x + \Delta$ to $x + 2\Delta$ (green), and $x + 2\Delta$ to $x + 3\Delta$ (red). **d**, Linear relationships among the pseudo model ensembles (Ensemble LR, colored lines) between LAI sensitivity to $CO_2$ of the three past periods and $\Delta$GPP from the projected period. Colored dots mark different models and the dashed lines represent associated pseudo observations for respective historical period. Yellow solid line depicts the constant Emergent Constraint on projected $\Delta$GPP irrespective of the past period .



1 **Table A1.** Overview of four possible cases of interaction between forcing, non-observable and observable identified in the Gedankenexperi-

2 ment: All linear, all nonlinear, and two mixed cases.

| Different assumptions | $\dfrac{\text{d[non}-\text{observable]}}{\text{d[forcing]}}$, e.g. $\dfrac{\text{d[GPP]}}{\text{d[CO}_2]}$ | | $\dfrac{\text{d[observable]}}{\text{d[non}-\text{observable]}}$, e.g. $\dfrac{\text{d[LAI]}}{\text{d[GPP]}}$ | |
|---|---|---|---|---|
| 1 | linear | | linear | |
| 2 | nonlinear | | linear | |
| 3 | linear | | nonlinear | |
| 4 | nonlinear | | nonlinear | |