# Peer review of "Investigating the Applicability of Emergent Constraints"

_Earth System Dynamics, 2018_

## Referee Comment (RC1) · Anonymous Referee #1 · 13 Jan 2019

The large disagreement of projections of future net land-atmosphere $CO_2$ flux in Earth-system models is the biggest uncertainty in future climate projections (Arora et al., 2013; Friedligstein et al. (2014). To tackle this issue, the application of emergent constraints (EC) to different carbon-cycle and ecosystem processes to reduce the range of the future land-sink estimates has become increasingly popular (Cox et al., 2013; Wenzel et al., 2014; Mystakidis et al., 2016; Wenzel et al., 2016). In this study Winkler et al. discuss the reasoning behind the application of EC in Earth-system modelling. They point to potential limitations, such as the need to accurately measure

the predictor and to find a robust relationship between predictor and predictand, and how that might change over time. They then use the sensitivity of Leaf-Area Index (LAI) to CO2 and temperature to constrain future estimates of Gross Primary Productivity in the Northern High-Latitudes.

In my opinion, the theoretical examination of the EC framework, sources of uncertainty and its limitations is particularly noteworthy and useful for the community (discussion around Figures 1, 4 and 6). I find the manuscript in the present form rather strenuous to read, without a fluid structure, several repetitions and sometimes omissions and inconsistencies that generate confusion. This can easily be improved during the revision: my suggestion would be to have a complete conceptual part discussing uncertainties and complications of the EC method before moving to the analysis of LAI data. There are, however, other points of this study that I find more problematic, and that need consideration before I can recommend its publication. I first describe my general concerns, and then include more specific comments for your consideration.

The introduction delves into the assumptions underlying the EC, different studies using EC to constrain the carbon-cycle sensitivity to global change and their limitations and uncertainties. I find that the introduction is missing a motivation statement that explains: (i) the need for the conceptual study presented here; (ii) why did the authors focused on the relationship between LAI and GPP (more on this below); (iii) the rationale behind the choice of trying to constrain $\triangle$GPP in the NHL only, since models that do well at simulating the effect of boreal/temperate ecosystem CO2 fluxes do not necessarily constrain better the global terrestrial sink (Schimel et al. 2015, Figure 3). The description of Winkler et al. (p2, l25 – p3, l2) is partly (but with less detail) described in the methods. I suggest mentioning here just the relevant aspects of their study. However, from this paragraph, it seems that one of the main conclusions of this manuscript is also an outcome of Winkler et al. (2018) – I mean the values of 3.4 $\pm$ 0.2 PgC.yr$^{-1}$ which are then presented again in the results section. This leaves me wondering to which extent is this study original, compared to that in revision in Nature

Communications. It's important that the authors clarify this, at least in their reply to the comments.

In the Methods section, the authors state that they "revisit the study of Winkler et al. (2018)" and "largely follow the methodology detailed in Winkler et al. (2018).". However, the reviewers (and potential readers) do not have access to this study to evaluate the methodology in detail nor to understand what exactly is being revisited. Moreover, that companion paper is not yet accepted for publication. Therefore, the authors should at least describe the methodology in more detail.

This is especially the case for the calculation of $\omega$, which is then used for a big part of the analysis of LAImax drivers. You explain that a PCA is performed on both variables ($CO_2$ and GDD0) to derive a proxy time-series that summarizes the evolution of both variables. The PCA is indeed suitable for such type of analysis and is probably better than multiple linear regressions used in other studies (e.g. Zhu et al., 2016). However, the authors give very little information about this crucial step of the analysis: is the PCA performed at pixel level, or for the large-scale aggregated values? What components do they retain from the PCA? (I'm assuming only PC1 is retained) What fraction of the variance does it explain? How does it relate to GDD0 and $CO_2$? How does it vary over time? Here, a plot showing $\omega$ over time would be very helpful. Moreover, the authors should keep in mind that $\omega$ does not "represent the overall forcing" (p9, l8-9), but only $CO_2$ and temperature.

The authors correctly state that one requirement of the EC method is that "a physically (or physiologically) based correlation between inter-model variations in an observable entity of the contemporary climate system (predictor) and a projected variable (predictand)" (p2, l26-27) exists. I find it, therefore, striking, that the authors do not discuss in any way why should LAI be used as a predictor of the CO2 fertilization

effect on GPP, and whether the linearity between the two variables in ESMs holds true for observations. Experimental CO2 enrichment studies did not find a direct effect between CO2 fertilization and increase in LAI (e.g. Körner et al., 2005) and LAI seems to increase non-linearly with increasing CO2 (Norby et al., 2005). Moreover, Norby et al. (2010) found strong influence of nutrient availability/limitation (not simulated in most CMIP5 ESMs) in the CO2 fertilization effect on ecosystem productivity, possibly because of mycorrhizal effect (Terrer et al., 2016). DeKawe and Medlyn (2014) have also shown that under increasing CO2, allocation of carbon to leaves decreased, rather than increasing (as implicitly assumed here), which was not well simulated by DGVMs. The link between CO2 fertilization, LAI and GPP is further complicated by how models simulate mortality and disturbances.

I understand that the authors have a stronger background on earth-system modelling and I would not expect them to make a full case on the relationships between CO2 fertilization, LAI and GPP. However, since they describe so well the need for a physical basis to the EC, they need to explain the choice of LAI as a predictor of future GPP (i.e. evidence for a mechanistic link), and whether the land-surface models composing the ESMs are able or not to correctly simulate the relevant processes for this relationship (see also Smith et al., 2016). In the current version of the manuscript, the authors do not make a strong case for their choice, and there is limited evidence (mostly from model-based studies to the best of my knowledge) to suggest that LAI sensitivity to CO2 can be a suitable predictor of future GPP. The authors could, for example, combine their analysis of $LAI_{max}$ sensitivity to CO2 and temperature with GPP changes estimated from observation-based datasets (e.g. FLUXCOM).

Specific comments:

P1, L 2: "promising results" of what?

P1, L3: What do you mean by "difficult to measure variable [...] at a potential future"? If you are trying to estimate a future state of a variable, it is by definition non-measureable?

P1, L7: "greening sensitivity to the CO2 forcing" ... but also temperature, right? (Methods).

P1, L18: Is the value of the GPP enhancement from this study or from Winkler et al. (in revision)?

P2, L4: "can have substantial uncertainties" → remove can. They have.

P2, L8: I'd move the "large-scale climate modes" to the paragraph about natural variability a few lines below.

P2, L12: "aims is to explore" → "aims to explore"

P2, L21: "namely, AS a method..."

P2, L24: In theory, could another relationship (non-linear) be used?

P2, L27: what do you mean by difficult to observe? Cox et al. (2013) used two variables that are relatively well observed (CO2 growth rate and tropical temperature).

P2, L32: What do you mean by "confirmed"?

P3, L17: "2xCO$_2$ world": you mean in model simulations, not in CO$_2$ enrichment experiments, right?

P6, L2: Here you mention that you also use precipitation to derive $\omega$, however later you mention only CO$_2$ and GDD0 were used. If you don't use, can you justify the exclusion of precipitation (non-significant trends? Non-significant effects?)

P7, L4-5: can you provide any lines of evidence to justify the assumption (non-model based).

P8, L4-5: What do you mean by "difficult to measure"? It's already repeated 2 times before.

P8, L6-9: What evidence do you provide for this? CO$_2$ enrichment experiments contradict this assumption.

P8, L15: "large area" → "large-scale"?

P8, L16-32: This is somewhat confusing since up until now you mention that you will analyse NHL. Please reformulate before in other to make clear that first you look at global values, and then focus on NHL (and provide justification to do so).

P8, L19-21: How much does GDD0 contribute to $\omega$ in the tropics? Can the low sensitivities in the tropics be due to your choice of temperature variable? I do not expect GDD0 to be a relevant temperature variable in the tropical band...

P9, L2-3: Indeed, but perhaps this is because of your inadequate choice of predictor for temperature (GDD0, rather than annual T, or some other metric)?

P9, L8-9: not the overall forcing, just two components of the forcing (CO2 and temperature). Please show the time-series of $\omega$.

P9, L17: "all pixels": of the globe, or just NHL?

P9, L26-29: Where do you show the corresponding increase in plant productivity? Where can I see that the distribution is approximately the same for the two variables? And if you have this data, where do you get GPP from, models or observations? Can you plot the GPP distribution for the same choice of pixels?

P10, L3-8: Is this also valid for ESM outputs?

P10, L19: What do you mean by "LAImax sensitivity cannot be accurately estimated irrespective of the window length".

P10, L20-21: Do you mean the signal to noise ratio of $\omega$? Unfortunately you don't show the time-series, so it's hard to follow.

P10, L23-26: But, in theory, that's the aim of the EC method. Do you mean that before considering using a given EC, one should evaluate the stability of the sensitivities?

P10, L29-30: It's not really shown in Figure 4.

P11, 4-6: Very good way to pose the question. But can you answer this in a pure model world? I'm not fully convinced.

P11, L8-10: Before (and after) you always use 7 models. It's not clear which model set is being used for which analysis. Are you using only 3 models to constrain future GPP changes? This does not seem consistent with Figure 6.

P11, L11-18: Not that surprising since all models are based in some way or another in the Farquhar photosynthesis model, which for the ppm ranges of $1xCO_2$ and $2xCO_2$ can possibly be approximated by a linear function, and in DGVMs the allocations schemes to leaves are strongly coupled to GPP (e.g. models don't simulate well non-structural carbon reserves, or changes in allocation)? Also, if models prescribe fixed LAImax (as some do), then this will strongly depend on the chosen model parametrization.

P11, L18: Why not call it simply "thought experiments" or "conceptual experiments", for non-german readers?

P11, L21: What do you mean by LAI? Annual values? Growing-season average? And why not LAImax?

P11, L24: "... responses" → add something like "of GPP to CO2 and of LAI to GPP" for clarity.

P11, L26 – P12, L5: Why did you choose Scenario 3? Scenario 4 in Figure A2 is much more plausible (GPP saturating for high levels of $CO_2$ because of basic physiology (Farquhar)).

P12, L9-10: "timing of saturation": where can we see this?

P12, L20: what LAImax are you referring to here? I assume you used AVHRR, since you explained (well) why MODIS is not suitable. But you need to clarify.

P12, L24-26: in the model world. You need to discuss whether observations support this.

P12, L26: I assume you mean "LAImax sensitivities" to $\omega$. Is this simulated $\omega$ or $\omega$ from observations? Over which period? If it is simulated $\omega$ you need to show how $\omega$ from historical simulations compares with $\omega$ from observations.

P13, L14: do models simulate compositional changes in these simulations? I.e. do they all include dynamic vegetation changes?

P13, L34: But observations seem to point out that climate change (warming and drying) probably cancels out the $CO_2$ fertilization effect (Penuelas et al., 2017), because of processes not well simulated by CMIP5 models - climate extremes, particularly heatwaves, mortality, disturbance – and further reinforced by nutrient limitations (also not simulated by most CMIP5 models).

P14, L8-9: Can you provide references for this?

P15, L2: is this an original result from this manuscript or from Winkler et al. in revision?

References

Arora et al., Journal of Climate (2013) 26, doi: 10.1175/JCLI-D-12-00494.1

Cox, et al. Nature (2013) 494.7437, 341.

DeKawe and Medlyn (2014), New Phytrologist, 203, 883-899.

Friedlingstein et al., Journal of Climate (2014) 27, doi: 10.1175/JCLI-D-12-00579.1

Körner et al. (2005), Science, 309, 5739.

Norby et al. (2005), PNAS, 102,50.

Mystakidis et al. Global Change Biology (2016) 22, 2198–2215, doi: 10.1111/gcb.13217

Schimel et al. PNAS (2015), 12, 2.

Smith et al. (2016) Nature Climate Change, 6, doi: 10.1038/NCLIMATE2879

Terrer et al. (2016) Science, 353, 6294.

Wenzel et al., J. Geophys. Res. Biogeosci., 119, 94–807, doi:10.1002/2013JG002591

Wenzel et al., Nature (2016), 538, 499-501.

Peñuelas et al. (2017) Nature Ecology and Evolution, 1, 1438-1445, doi: 10.1038/s41559-017-0274-8

Zhu et al. (2016) Nature Clim. Change, 6, 791–795.

---

## Author Comment (AC1) · 6 Mar 2019

**Authors' Response to Referee 1 (ESDD esd-2018-71)**

March 6, 2019

*The large disagreement of projections of future net land-atmosphere $CO_2$ flux in Earth-system models is the biggest uncertainty in future climate projections (Arora et al., 2013; Friedlingstein et al., 2013). To tackle this issue, the application of emergent constraints (EC) to different carbon-cycle and ecosystem processes to reduce the range of the future land-sink estimates has become increasingly popular (Cox et al., 2013; Wenzel et al., 2014; Mystakidis et al., 2016; Wenzel et al., 2016). In this study Winkler et al. discuss the reasoning behind the application of EC in Earth-system modelling. They point to potential limitations, such as the need to accurately measure the predictor and to find a robust relationship between predictor and predictand, and how that might change over time. They then use the sensitivity of Leaf-Area Index (LAI) to $CO_2$ and temperature to constrain future estimates of Gross Primary Productivity in the Northern High-Latitudes. In my opinion, the theoretical examination of the EC framework, sources of uncertainty and its limitations is particularly noteworthy and useful for the community (discussion around Figures 1, 4 and 6). I find the manuscript in the present form rather strenuous to read, without a fluid structure, several repetitions and sometimes omissions and inconsistencies that generate confusion. This can easily be improved during the revision: my suggestion would be to have a complete conceptual part discussing uncertainties and complications of the EC method before moving to the analysis of LAI data. There are, however, other points of this study that I find more problematic, and that need consideration before I can recommend its publication. I first describe my general concerns, and then include more specific comments for your consideration.*

We thank the reviewer for her/his detailed and very constructive review of our manuscript. We appreciate that the reviewer finds our study particularly noteworthy and useful for the community, but we also notice the reviewer's concerns. All revisions done in response to the reviewer's comments improved the structure and overall readability of the manuscript.

**1 General Comments**

*1.1 The introduction delves into the assumptions underlying the EC, different studies using EC to constrain the carbon-cycle sensitivity to global change and their limitations and uncertainties. I find that the introduction is missing a motivation statement that explains: (**i**) the need for the conceptual study presented here; (**ii**) why did the authors focused on the relationship between LAI and GPP (more on this below); (**iii**) the rationale behind the choice of trying to constrain $\Delta GPP$ in the NHL only, since models that do well at simulating the effect of boreal/temperate ecosystem $CO_2$ fluxes do not necessarily constrain better the global terrestrial sink (Schimel et al. (2015), Figure 3).*

We agree, that the introduction lacks a clear motivation why a conceptual approach to the EC method is needed. In recent years, many studies have been published applying the EC method to constrain essential entities of the Earth system. The method will become even more popular with the upcoming CMIP6 model simulations. However, the literature is missing a detailed description of the applicability and limitations of the EC method, resulting in a somewhat arbitrary application and methodological inconsistencies among various studies. To account for that, this conceptual study is needed, which elaborates on the behavior of the EC method (**i**); There is no specific reason why we based our conceptual study on the relationship between LAI and GPP. We adduced these variables to build a case study in order to scrutinize the EC method. In theory, the results are qualitatively transmissive to other sets of predictors and predictands (**ii**); We focused our analysis on the northern high latitudes (NHL) because of two reasons. First, ecosystems in NHL are barely influenced by human land use. Thus, the changes of vegetation greenness

are natural responses to the forcing rather than agricultural artifacts. Second, independent remote sensors (AVHRR, MODIS) yield comparable greening sensitivities for the NHL, although the MODIS time series is yet too short to derive a statistical robust estimate (**iii**). We edited the introduction in the revised manuscript to account for the these comments.

*1.2 The description of Winkler et al. (P2, L25 - P3, L2) is partly (but with less detail) described in the methods. I suggest mentioning here just the relevant aspects of their study.*

In the revised manuscript, we added a description of the methodology in Winkler et al. (2019), especially, we make abundantly clear how $\omega$ is derived and describe its characteristics for models and observations. Please see also responses to comment 1.5 and 2.26.

*1.3 However, from this paragraph, it seems that one of the main conclusions of this manuscript is also an outcome of Winkler et al. (2018) – I mean the values of 3.4 ± 0.2 Pg C yr$^{-1}$ which are then presented again in the results section. This leaves me wondering to which extent is this study original, compared to that in revision in Nature Communications. It's important that the authors clarify this, at least in their reply to the comments.*

Winkler et al. (2019) present constraints on projected future plant productivity in NHL using greening sensitivity as well as independent observational resources such as ground-measurements of $CO_2$ and atmospheric $CO_2$ inversion products. The study in hand focuses on the concept of EC, its applicability and limitations, building on the EC presented in Winkler et al. (2019). We agree that presenting the $\Delta$GPP constraint of 3.4 ± 0.2 Pg C yr$^{-1}$ as key result in both studies, is problematic. We address this issue in the revised manuscript. Please see also our response to comment 2.4.

*1.4 In the Methods section, the authors state that they "revisit the study of Winkler et al. (2018)" and "largely follow the methodology detailed in Winkler et al. (2018)". However, the reviewers (and potential readers) do not have access to this study to evaluate the methodology in detail nor to understand what exactly is being revisited. Moreover, that companion paper is not yet accepted for publication. Therefore, the authors should at least describe the methodology in more detail.*

Yes, we acknowledge that access to the companion article is needed to better comprehend the methodology in this study. The article by Winkler et al. (2019) is now published and openly available on the website of *Nature Communications* (https://rdcu.be/bpELU). We included a more comprehensive methods section in the revised manuscript. Please see also responses to comments 1.2 and 1.5.

*1.5 This is especially the case for the calculation of $\omega$, which is then used for a big part of the analysis of LAI$_{max}$ drivers. You explain that a PCA is performed on both variables ($CO_2$ and GDD0) to derive a proxy time-series that summarizes the evolution of both variables. The PCA is indeed suitable for such type of analysis and is probably better than multiple linear regressions used in other studies (e.g. Zhu et al. (2016)). However, the authors give very little information about this crucial step of the analysis: is the PCA performed at pixel level, or for the large-scale aggregated values? What components do they retain from the PCA (I'm assuming only PC1 is retained)? What fraction of the variance does it explain? How does it relate to GDD0 and $CO_2$? How does it vary over time? Here, a plot showing $\omega$ over time would be very helpful. Moreover, the authors should keep in mind that $\omega$ does not "represent the overall forcing" (P9, L8-9), but only $CO_2$ and temperature.*

We agree, that a more detailed description of the derivation of $\omega$ needs to be provided. PCA was performed on large-scale aggregated values as well as on pixel level to investigate on spatial variations. We only retain the first principal component (denoted $\omega$), which explains a large fraction of the variance, ranging approximately from 70% ot 90% in models and observations (for more details see Table R1-1, also included in Supplementary Information in the companion article). Figure R1-1 depicts the temporal development of $CO_2$ and GDD0 as well as the principal component $\omega$ for observations. This figure, with some modifications, has been included in the appendix of the revised manuscript. Yes, we acknowledge that $CO_2$ and GDD0 do not represent the overall historical forcing, but we assume that these are the main drivers causing observed changes in the NHL region. Please see also response to comment 1.2.

[Figure]

Figure R1- 1: Standardized temporal anomalies of annual averaged atmospheric $CO_2$ concentration, area-weighted averaged GDD0 for NHL, and their leading principal component $\omega$ in observations.

Table R1- 1: Summary data for Principal Component Analysis and $LAI_{max}$ sensitivity estimation.

| Model | Explained variance by $\omega$ | $LAI_{max}$ sensitivity to $\omega$, (m$^2$ m$^{-2}$ unit $\omega$) | Correlation coefficient |
|---|---|---|---|
| MIROC-ESM | 0.89 | $0.049 \pm 3.3e\text{-}3$ | 0.93 |
| CESM1-BGC | 0.83 | $0.014 \pm 1.4e\text{-}3$ | 0.86 |
| GFDL-ESM2M | 0.64 | $0.022 \pm 3.2e\text{-}3$ | 0.76 |
| CanESM2 | 0.91 | $0.013 \pm 1.0e\text{-}3$ | 0.91 |
| HadGEM2-ES | 0.94 | $0.075 \pm 3.5e\text{-}3$ | 0.97 |
| MPI-ESM-LR | 0.77 | $0.028 \pm 1.8e\text{-}3$ | 0.94 |
| NorESM1-ME | 0.84 | $0.0088 \pm 0.8e\text{-}3$ | 0.88 |
| Observations | 0.9 | $0.045 \pm 6.4e\text{-}3$ | 0.78 |

*1.6 The authors correctly state that one requirement of the EC method is that "a physically (or physiologically) based correlation between inter-model variations in an observable entity of the contemporary climate system (predictor) and a projected variable (predictand)" (P2, L26-27) exists. I find it, therefore, striking, that the authors do not discuss in any way why should LAI be used as a predictor of the $CO_2$ fertilization effect on GPP, and whether the linearity between the two variables in ESMs holds true for observations. Experimental $CO_2$ enrichment studies did not find a direct effect between $CO_2$ fertilization and increase in LAI (e.g. Körner et al., 2005) and LAI seems to increase non-linearly with increasing $CO_2$ (Norby et al. (2005)). Moreover, Norby et al. (2010) found strong influence of nutrient availability/limitation (not simulated in most CMIP5 ESMs) in the $CO_2$ fertilization effect on ecosystem productivity, possibly because of mycorrhizal effect (Terrer et al. (2016))). ? have also shown that under increasing $CO_2$, allocation of carbon to leaves decreased, rather than increasing (as implicitly assumed here), which was not well simulated by DGVMs. The link between $CO_2$ fertilization, LAI and GPP is further complicated by how models simulate mortality and disturbances.*

The link between LAI, GPP, and elevated $CO_2$ concentration is a complicated subject matter, as the referee thoroughly describes. In terms of *in-situ* measurements, there is no clear picture emerging. Körner et al. (2005) finds no significant coupling between elevated $CO_2$ and increased LAI in a Swiss forest site for a study period of four years. Norby et al. (2005) analyzed measurements from four different FACE experiments in the northern mid-latitudes (USA and Italy). They detect a non-linear relationship between increasing $CO_2$ and LAI. However, non-linearity is to be expected for such a sharp increase of $CO_2$ concentration (quasi-instant forcing of 174 ppm) and is not

comparable to the real-world response (annual forcing of 2-3 ppm). Please see also our response to comment 2.14 for a more detailed discussion on the assumption of linearity in the relationship between LAI and $CO_2$ for the last decades. Norby et al. (2005) also report, that their analysis suggested that at low LAI, elevated $CO_2$ was causing structural changes and substantial increase in absorbed photosynthetic active radiation, in general agreement with satellite measurements of low LAI regions, especially in NHL. De Kauwe et al. (2014) analyzed measurements from two FACE experiments located in North America (North Carolina and Tennessee, USA). They find that specific leaf area (SLA, the ratio of leaf area to leaf mass) decreased, but report a general increase in LAI as response to elevated $CO_2$.

In general, conclusions drawn from FACE experiments, owed to their setup, are not representative for long-term observed changes on ecosystem-scale. We agree with the referee, that the current manuscript lacks an in-depth discussion on the causal link between predictor and predictand. However, this aspect is discussed in more detail in the companion paper by Winkler et al. (2019) and illustrated in Supplementary Figure 1 - *Schematic of the Emergent Constraint concept.* In the revised manuscript we discuss in more detail the causal link between predictor and predictand. Please see also our response to comment 2.41.

*1.7 I understand that the authors have a stronger background on earth-system modelling and I would not expect them to make a full case on the relationships between $CO_2$ fertilization, LAI and GPP. However, since they describe so well the need for a physical basis to the EC, they need to explain the choice of LAI as a predictor of future GPP (i.e. evidence for a mechanistic link), and whether the land-surface models composing the ESMs are able or not to correctly simulate the relevant processes for this relationship (see also Kolby Smith et al. (2016)). In the current version of the manuscript, the authors do not make a strong case for their choice, and there is limited evidence (mostly from model-based studies to the best of my knowledge) to suggest that LAI sensitivity to $CO_2$ can be a suitable predictor of future GPP. The authors could, for example, combine their analysis of $LAI_{max}$ sensitivity to $CO_2$ and temperature with GPP changes estimated from observation-based datasets (e.g. FLUXCOM).*

The referee makes a very good proposal in analyzing other observation-based datasets to corroborate the EC estimate. We already conducted such analysis and is part of Winkler et al. (2019). Amongst other data resources, we analyzed all available FLUXCOM datasets of upscaled eddy covariance flux measurements for NHL GPP. However, these datasets were designed not to capture long-term changes as well as interannual variability, and thus, cannot be applied for a temporal analysis (e.g. Anav et al., 2015). But one can build on the spatial information to investigate the correlation between $LAI_{max}$ and GPP. Using the climatologic mean of the recommended ensemble median of all FLUXCOM datasets and two independent sets of satellite observed LAI, we find a striking linear relationship for the northern high latitudes (Figure R1-2a and b). This tight linear relation between the two variables over a wide range of values suggests that changes in GPP also result in changes in $LAI_{max}$. In general, model simulations and large-scale observational datasets clearly indicate that LAI sensitivity to $CO_2$ ($\omega$ for temperature-limited ecosystems) is a suitable predictor of GPP for increasing $CO_2$ forcing in the NHL.

Further, we assess the relationship between changes in GPP and LAI exclusively using *in-situ* flux measurements, although these records are yet to short for a statistically robust analysis. We selected the longest FluxNet time series existing for the NHL, Hyytiala, Finland (61.8474° N, 24.2948° E, 1996 - 2014). We took the surrounding pixels of the long-term but rather coarse (AVHRR, 1/12°, 1982 - 2016) as well as short-term but higher resolution (MODIS, 500m, 2000 - 2016) satellite observations of LAI. We find contemporary trends in GPP and LAI, but the linear relation between the *in-situ* measured GPP and long-term AVHRR satellite datasets is rather weak due to the coarse resolution. Thus, to match the flux tower footprint, we have to make recourse to high resolution satellite observations of MODIS. MODIS LAI and AVHRR LAI (both analyzed in our study) have strong correlation and the latest AVHRR LAI datasets were developed by referencing to MODIS LAI (Zhu et al., 2013). For the MODIS time-series we find a much stronger relationship to the flux measurements and therefore confirms the tight connection between changes in GPP and LAI for the site in Hyytiala, Finland (Figure R1-2c). However, the overlapping period

of MODIS and FluxNet is yet too short to derive reliable estimates. Anav et al. (2015) also analyzed other eddy covariance flux measurement sites and find a general agreement on increasing GPP.

[Figure]

Figure R1- 2: Strong correlation in the climatologic mean in observational datasets between $LAI_{max}$ derived from two independent satellite sensors, MODIS (**a**) and AVHRR (**b**), and the ensemble median annual average GPP from the FLUXCOM ensemble for the northern high latitudes. Color density indicates the probability distribution estimated using Gaussian kernel. **c**, Contemporary trends in the longest *in-situ* GPP flux measurement record in the NHL and the study site surrounding pixels of high resolution LAI satellite observations. The blue line shows the best linear fit and the shading shows the 95% confidence interval.

**2   Specific comments:**

**2.1** *P1, L 2: "promising results" of what?*

This sentence has been rewritten to be more specific.

**2.2** *P1, L3: What do you mean by "difficult to measure variable [...] at a potential future"? If you are trying to estimate a future state of a variable, it is by definition non-measurable?*

The statement 'difficult-to-measure' only refers to 'variable' and not to 'a potential future'. We rewrote this paragraph to avoid misunderstandings. Please see also response to comment 2.15

**2.3** *P1, L7: "greening sensitivity to the $CO_2$ forcing" ... but also temperature, right? (Methods).*

We investigate both types of sensitivity, so, the greening response to rising $CO_2$ as well as to the combined signal of rising $CO_2$ and temperature (GDD0). The latter approach is necessary for temperature-limited ecosystems and is only applied in the analysis focusing on NHL.

**2.4** *P1, L18: Is the value of the GPP enhancement from this study or from Winkler et al. (in revision)?*

This result is a subject in both studies, however, we discuss it with different perspectives. In the revised manuscript, we define the focus of this study more precisely to avoid such misunderstandings. Please see also response to comment 1.3.

**2.5** *P2, L4: "can have substantial uncertainties" ? remove can. They have.*

We agree. The sentence was modified.

**2.6** *P2, L8: I'd move the "large-scale climate modes" to the paragraph about natural variability a few lines below.*

The sentence should give an overview of the range of aspects which are underrepresented in ESMs, from local short time-scale extreme events to long-term large-scale climatic modes. Hence, we prefer not to modify this section.

We corrected the sentence.

*2.8 P2, L21: "namely, AS a method. . ."*

We agree, the sentence reads better now.

*2.9 P2, L24: In theory, could another relationship (non-linear) be used?*

Yes, in theory, a non-linear relationship between predictor and predictand in an multi-model ensemble is conceivable, but this requires a reasonable process-based justification. For instance, there are attempts to establish an EC using an exponential relationship between historical warming (predictor) and equilibrated temperature increase for a doubling of atmospheric $CO_2$ (predictand, equilibrium climate sensitivity, ECS). This approach implies that models with strong historical warming should predict a disproportional high ECS. To build a reliable EC, one has to identify the process causing this disproportionality in the model ensemble.

*2.10 P2, L27: what do you mean by difficult to observe? Cox et al. (2013) used two variables that are relatively well observed ($CO_2$ growth rate and tropical temperature).*

Cox et al. (2013) used variations in the observables $CO_2$ growth rate and tropical temperature to constrain land carbon storage in the tropics, the latter being the variable that is difficult to observe.

*2.11 P2, L32: What do you mean by "confirmed"?*

The relationship between snow-albedo feedback strength of the current seasonal cycle and projected feedback to long-term warming has been detected in the CMIP3 ensemble and also exists in the CMIP5 ensemble. If an "Emergent Relationship" is independent of the analyzed model ensemble, it is considered as 'confirmed' in the EC literature.

*2.12 P3, L17: "2xCO₂ world": you mean in model simulations, not in $CO_2$ enrichment experiments, right?*

Yes, '2xCO₂ world' refers to model simulations. We rewrote this section to be more specific.

*2.13 P6, L2: Here you mention that you also use precipitation to derive $\omega$, however later you mention only $CO_2$ and GDD0 were used. If you don't use, can you justify the exclusion of precipitation (non-significant trends? Non-significant effects?)*

Precipitation and temperature are used to derive climatic regimes (see Figure 2 in the manuscript). For each climatic regime, we derive the greening sensitivity to $CO_2$. First, only the sensitivities to rising atmospheric $CO_2$ concentration are calculated to obtain comparability between the different climatic regimes, vegetation classes, and latitudinal bands (see Figure 2 in the manuscript). Then, we focus on the northern high latitudes, where we also have to take temperature into account and derive the greening sensitivity to $\omega$, the combined signal of $CO_2$ and GDD0. We rewrote this passage to be more clear on that matter.

*2.14 P7, L4-5: can you provide any lines of evidence to justify the assumption (non-model based).*

The increase of observed $CO_2$ concentration (annual average) throughout the satellite era can be considered as quasi-linear (Figure R1-3). Our analyses of remote sensing datasets of LAI from different sources (AVHRR, MODIS) also suggest linearly increasing trends. This finding was also reported by several preceding studies (Zhu et al., 2016; Mao et al., 2016; Forkel et al., 2016; Mahowald et al., 2016). Forkel et al. (2016, Fig. 1 - *Amplification of plant activity in the northern biosphere*) analyzed several observational datasets for the northern ecosystems of the last 30 to 40 years and report evidence for a linearly changing system. The bottom line is, there is no observational indication of a non-linear relationship between LAI and $CO_2$, at least not for the $CO_2$ forcing of the last decades (from ∼340 to ∼410 ppm.)

[Figure]

Figure R1- 3: A quasi-linear increase in observations of global monthly mean $CO_2$ concentrations since 1980; image taken from https://www.esrl.noaa.gov/gmd/ccgg/trends/gl_full.html, February 14, 2019.

**2.15** *P8, L4-5: What do you mean by "difficult to measure"? It's already repeated 2 times before.*

The concept of Emergent Constraints is to constrain an entity (predictand) of the Earth system that is not-directly or not-at-all observable (e.g. at a potential future state). This can be achieved by using an observable that is physically connected to the predictand. We understand the confusion about the term 'difficult-to-measure' with regard to projected estimates of GPP. To be more clear on that matter, we modified the terminology in the revised manuscript. Please see also response to comment 2.2.

**2.16** *P8, L6-9: What evidence do you provide for this? $CO_2$ enrichment experiments contradict this assumption.*

Please see responses to comments 1.6 and 2.41.

We agree.

**2.18** *P8, L16-32: This is somewhat confusing since up until now you mention that you will analyse NHL. Please reformulate before in other to make clear that first you look at global values, and then focus on NHL (and provide justification to do so).*

First, we present the observable on global scale aggregated for different climatic regimes, vegetation types, and latitudinal bands. Then, we show that LAI is only a meaningful predictor for changes in GPP in the northern high latitudes. We rewrote this section and provide better justification for our approach.

**2.19** *P8, L19-21: How much does GDD0 contribute to $\omega$ in the tropics? Can the low sensitivities in the tropics be due to your choice of temperature variable? I do not expect GDD0 to be a relevant temperature variable in the tropical band. . .*

Yes, we agree, GDD0 is not a relevant temperature variable for the tropical regions. We only consider GDD0 in the NHL, as part of $\omega$. When deriving sensitivities for global comparison (e.g. comparing tropical, mid-latitude, and high-latitude sensitivities; Figure 2 in the manuscript), we only consider the signal of rising $CO_2$ concentration and neglect temperature. Thereafter, we focus on the NHL, because there we obtain a clear LAI signal, e.g. not being distorted by human land use. We derive LAI sensitivity to $\omega$, so, also accounting for temperature and its variations, an important aspect for temperature-limited ecosystems. Also LAI in the tropics is quite sensitive to temperature variations, particularly to anomalies associated to ENSO. Thus, for a study focused on the tropics one should also consider temperature in estimating LAI sensitivities.

*2.20 P9, L2-3: Indeed, but perhaps this is because of your inadequate choice of predictor for temperature (GDD0, rather than annual T, or some other metric)?*

No, this is not the case. Please see response to comment 2.19. We only take temperature into account when focusing on NHL.

*2.21 P9, L8-9: not the overall forcing, just two components of the forcing ($CO_2$ and temperature). Please show the time-series of $\omega$.*

Please see our response to comment 1.5, Table R1-1 and the time-series of $\omega$ in Figure R1-1.

*2.22 P9, L17: "all pixels": of the globe, or just NHL?*

In the manuscript, we state "We focus further analyses on the NHL region [...]" (P9, L2). Hence, we only show global comparison of LAI sensitivities to $CO_2$ in section 3.1 and, thereafter, we concentrate on LAI sensitivity to $\omega$ in NHL. We modified this section to be more precise.

*2.23 P9, L26-29: Where do you show the corresponding increase in plant productivity? Where can I see that the distribution is approximately the same for the two variables? And if you have this data, where do you get GPP from, models or observations? Can you plot the GPP distribution for the same choice of pixels?*

We use CMIP5 model output to show that the distribution of pixels with significant changes of the predictor (LAI sensitivity to $\omega$, historical simulation) and the predictand (GPP, 1pctCO2) are approximately the same. Figure R1-4 compares respective distributions for all CMIP5 models. All models, except HadGEM2-ES, confirm that the pixels that show significant historical LAI sensitivity to $\omega$ are approximately also the pixels showing significant changes in GPP for 2×$CO_2$, resulting in similar distributions. Note, that the variables LAI and GPP had to be normalized for comparison in the same figure. This analysis is corroborating our statement in the manuscript, that averaging the equally distributed estimates does not affect the predictor-predictand relationship in the model ensemble (P9, L29-30). Also, the results shown in Anav et al. (2013, 2015) indicate spatial correlation of increasing GPP and LAI.

Long-term and large-scale changes in GPP still cannot be obtained form observations. Upscaled FluxNet measurements (i.e. FLUXCOM datasets) also rely on statistical models (e.g. neuronal networks) and are designed not to capture long-term changes (e.g. Anav et al., 2015). Thus, these datasets can only be applied for certain types of analyses, e.g. spatial patterns or natural variability. Please see also our response to comment 1.7. For completeness, we include a modified version of Figure R1-4 in the appendix of the manuscript.

*2.24 P10, L3-8: Is this also valid for ESM outputs?*

Yes, this statement is also valid for ESM simulation output. We discuss this aspect in the manuscript (P10, L27-30).

*2.25 P10, L19: What do you mean by "$LAI_{max}$ sensitivity cannot be accurately estimated irrespective of the window length".*

This statement refers to Figure 4 in the manuscript. Figure 4 shows $LAI_{max}$ sensitivity to $\omega$ for the historical period from 1860 to 2005 for different moving window lengths (15yr, 30yr, and 40yr). In the decades around the turn of the 20th century, $LAI_{max}$ sensitivity to $\omega$ is fluctuating from negative to positive numbers for all window lengths. This is, because $CO_2$ forcing is low, and thus, natural variability dominates. Under these circumstances, $LAI_{max}$ sensitivity to $\omega$ cannot be accurately estimated.

*2.26 P10, L20-21: Do you me an the signal to noise ratio of $\omega$? Unfortunately you don't show the time-series, so it's hard to follow.*

When $CO_2$ forcing is low, natural variability (*noise*) is dominating and influencing the estimation of $LAI_{max}$ sensitivity to $\omega$. But, when $CO_2$ forcing grows stronger, the LAI response (*signal*) is exceeding the noise and $LAI_{max}$ sensitivity to the forcing can be estimated. Please see also our responses to comment 1.5 and 2.25., Table R1-1 and the time-series of $\omega$ in Figure R1-1.

**2.27** *P10, L23-26: But, in theory, that's the aim of the EC method. Do you mean that before considering using a given EC, one should evaluate the stability of the sensitivities?*

This section is refereeing to the comparability of sensitivities in window length and location between observations and models. In other words, the observed and modeled predictors have to be obtained from the same point in time (level of $CO_2$ forcing) and comparable temporal window lengths, so, all predictors have to be representative for the same state of the system. Yes, besides evaluating the stability of the predictor, one has to evaluate the comparability of predictors. So, the aim of the EC method is to use these predictors to constrain an entity of interest (predictand) at a different state (forcing) of the system.

**2.28** *P10, L29-30: It's not really shown in Figure 4.*

We argue, that Figure 4 in the manuscript clearly shows that $LAI_{max}$ sensitivity estimation becomes more stable with strengthening forcing and increasing window length. Please compare different colored lines representing different window lengths and variability for different points in time, i.e. $CO_2$ concentration.

**2.29** *P11, 4-6: Very good way to pose the question. But can you answer this in a pure model world? I'm not fully convinced.*

In this section, we show that the EC method can be applied also when the underlying relationship between predictor and predictand is changing with increasing forcing (e.g. from linear to non-linear). Predictions of future GPP are based on our current understanding of the system. We expect that saturation will occur with increasing $CO_2$. In spite of this non-linear response, we illustrate that the EC relationship in the model ensemble can remain linear. From observations only, we cannot obtain ecosystem-scale estimates of GPP increase for a high $CO_2$-world. So, yes, we can and must answer this question in a pure model world.

**2.30** *P11, L8-10: Before (and after) you always use 7 models. It's not clear which model set is being used for which analysis. Are you using only 3 models to constrain future GPP changes? This does not seem consistent with Figure 6.*

Yes, we agree, this is confusing. In general, we use as many models as possible for the EC analysis (here, 7 models). In Figure 4 and 5 in the manuscript, we only show 3 of the 7 models, because all models show qualitatively the same. We selected these three models, because they span the full range of GPP predictions (CESM1-BGC: lowest estimate, HadGEM2-ES: highest estimate, and MIROC-ESM: closest to EC estimate). We generated two additional figures (shown in the appendix of the revised manuscript) which display the results of the other 4 models analogous to Figure 4 and 5 in the manuscript.

**2.31** *P11, L11-18: Not that surprising since all models are based in some way or another in the Farquhar photosynthesis model, which for the ppm ranges of 1xCO2 and 2xCO2 can possibly be approximated by a linear function, and in DGVMs the allocations schemes to leaves are strongly coupled to GPP (e.g. models don't simulate well non-structural carbon reserves, or changes in allocation)? Also, if models prescribe fixed $LAI_{max}$ (as some do), then this will strongly depend on the chosen model parametrization.*

Yes, we agree, it is not surprising that all models show saturation at higher $CO_2$ levels. However, here we make the point, that despite the expected non-linearity of the predictor-predictand relationship at higher $CO_2$ levels, the inter-model relationship in the ensemble space can remain linear. This is a somewhat counter-intuitive aspect of the EC method and essential for its interpretation.

**2.32** *P11, L18: Why not call it simply "thought experiments" or "conceptual experiments", for non-german readers?*

*Gedankenexperiment* is an universal scientific term such as the German word *Ansatz*.

**2.33** *P11, L21: What do you mean by LAI? Annual values? Growing-season average? And why not $LAI_{max}$?*

We intended to simplify the terminology in the *Gedankenexperiment*. Since the time dimension does not play a role in this conceptual framework, LAI expressed as annual average, growing season average, or annual maximum has no meaning. However, we acknowledge that the changed terminology can cause some confusion. Therefore, we stick to $LAI_{max}$ in the revised manuscript.

*2.34 P11, L24: ". . . responses" ? add something like "of GPP to $CO_2$ and of LAI to GPP" for clarity.*

We rewrote the sentence for clarity.

*2.35 P11, L26 - P12, L5: Why did you choose Scenario 3? Scenario 4 in Figure A2 is much more plausible (GPP saturating for high levels of $CO_2$ because of basic physiology (Farquhar)).*

We chose Scenario 3 to highlight the interplay of linear and non-linear relationships between forcing, predictor, and predictand. But we agree with the referee that Scenario 4 is the most plausible, which we also discuss in the manuscript (P12, L9-11).

*2.36 P12, L9-10: "timing of saturation": where can we see this?*

Figure 4 and Table 2 in the manuscript illustrate that the CMIP5 models show saturation of the relationship between $\Delta LAI_{max}$ and $\Delta GPP$ with increasing $CO_2$ forcing. The slopes in Figure 4 (detailed estimates in Table 2) reveal that the strength and 'timing' of saturation (i.e. at what level of $CO_2$ concentration) differs among the models. In the revised manuscript, we implemented more accurate description and references to tables and figures.

*2.37 P12, L20: what $LAI_{max}$ are you referring to here? I assume you used AVHRR, since you explained (well) why MODIS is not suitable. But you need to clarify.*

Yes, we used AVHRR data. We added more details to this section in the manuscript.

*2.38 P12, L24-26: in the model world. You need to discuss whether observations support this.*

Please see response to general comment 1.6.

*2.39 P12, L26: I assume you mean "$LAI_{max}$ sensitivities" to $\omega$. Is this simulated $\omega$ or $\omega$ from observations? Over which period? If it is simulated $\omega$ you need to show how $\omega$ from historical simulations compares with $\omega$ from observations.*

$LAI_{max}$ sensitivity to $\omega$ is calculated for observations and each model separately for approximately the same time period. Please see Table R1-1 for more details. This approach enables an accurate comparison between the simulated and observed predictor variables. Also, more details on $\omega$ can be looked up in the supplementary information for Winkler et al. (2019).

*2.40 P13, L14: do models simulate compositional changes in these simulations? I.e. do they all include dynamic vegetation changes?*

Yes, most of the models include dynamic vegetation. In the revised manuscript, we include a short description of the representation of dynamic vegetation in CMIP5 models. In general, the historical and idealized model setups of the CMIP5 land components are comprehensively explained in several studies, such as Wenzel et al. (2014); Mahowald et al. (2016); Arora et al. (2013); Winkler et al. (2019). This is why we refrain from providing a detailed overview of the CMIP5 models in this study.

*2.41 P13, L34: But observations seem to point out that climate change (warming and drying) probably cancels out the $CO_2$ fertilization effect (Peñuelas et al., 2017), because of processes not well simulated by CMIP5 models - climate extremes, particularly heatwaves, mortality, disturbance and further reinforced by nutrient limitations (also not simulated by most CMIP5 models).*

The referee addresses one of the key problems in current climate and carbon cycle research. On the one hand, we expect that $CO_2$ fertilization is causing enhanced plant growth based on our physiological understanding. Many studies find evidence for this expectation. Especially, the Global Carbon Project suggests that $\sim$30% of the anthropogenic $CO_2$ emissions are taken up the terrestrial biosphere (so, current land sink is $\sim$ 12 Pg C yr$^{-1}$, Quéré et al., 2018). On the other hand, observations (esp. on local scale) suggest that the net carbon uptake by plants for a higher $CO_2$ world is not changing due compensating effects (e.g. Peñuelas et al., 2017). Obviously, there is a paradox in place: Where do the 12 Pg carbon go every year, if plants do not take up more carbon with rising $CO_2$? Future research needs to address this issue in more depth. For the NHL, however, we find robust observational evidence (Keeling et al., 1996; Myneni et al., 1997; Graven et al., 2013; Forkel et al., 2016; Winkler et al., 2019) that carbon uptake by plants is increasing, which is the baseline for the study in hand.

**2.42 *P14, L8-9: Can you provide references for this?**

We assume the comment is referring to P15. There are several studies indicating that greening in the high northern latitudes is caused by indirect drivers associated to increasing $CO_2$, such as warming and $CO_2$ fertilization (Myneni et al., 1997; Forkel et al., 2016; Zhu et al., 2016). At high LAI regions, GPP might also increase due to $CO_2$ fertilization without an enhancement of LAI. In rural areas, the observed greening is mainly caused by direct drivers such irrigation, application of fertilizers, and double cropping as shown recently by Chen et al. (2019). We added the references in the manuscript.

**2.43 *P14, L2: is this an original result from this manuscript or from Winkler et al. in revision?**

We assume the comment is referring to P16. As we explained in response to comment 2.4, this result is subject in both studies, but discussed with different perspectives.

[Figure]

Normalized $x$-axis

Figure R1- 4: Similar pixel distributions of LAI sensitivity to $\omega$ (red, historical simulations) and temporal trends in GPP (blue, 1pctCO2, until $2\times CO_2$) for NHL. All CMIP5 models are shown. Only significant pixels are included (Mann-Kendall test, p < 0.1). To obtain comparability, the $x$-axis was normalized and has only qualitative meaning.

**References**

Anav, A., Friedlingstein, P., Beer, C., Ciais, P., Harper, A., Jones, C., Murray-Tortarolo, G., Papale, D., Parazoo, N. C., Peylin, P., Piao, S., Sitch, S., Viovy, N., Wiltshire, A., and Zhao, M. (2015). Spatiotemporal patterns of terrestrial gross primary production: A review. *Reviews of Geophysics*, 53(3):2015RG000483.

Anav, A., Friedlingstein, P., Kidston, M., Bopp, L., Ciais, P., Cox, P., Jones, C., Jung, M., Myneni, R., and Zhu, Z. (2013). Evaluating the Land and Ocean Components of the Global Carbon Cycle in the CMIP5 Earth System Models. *Journal of Climate*, 26:6801–6843.

Arora, V. K., Boer, G. J., Friedlingstein, P., Eby, M., Jones, C. D., Christian, J. R., Bonan, G., Bopp, L., Brovkin, V., Cadule, P., Hajima, T., Ilyina, T., Lindsay, K., Tjiputra, J. F., and Wu, T. (2013). Carbon–Concentration and Carbon–Climate Feedbacks in CMIP5 Earth System Models. *Journal of Climate*, 26:5289–5314.

Chen, C., Park, T., Wang, X., Piao, S., Xu, B., Chaturvedi, R. K., Fuchs, R., Brovkin, V., Ciais, P., Fensholt, R., Tømmervik, H., Bala, G., Zhu, Z., Nemani, R. R., and Myneni, R. B. (2019). China and India lead in greening of the world through land-use management. *Nature Sustainability*, 2(2):122.

Cox, P. M., Pearson, D., Booth, B. B., Friedlingstein, P., Huntingford, C., Jones, C. D., and Luke, C. M. (2013). Sensitivity of tropical carbon to climate change constrained by carbon dioxide variability. *Nature*, 494(7437):341–344.

De Kauwe, M. G., Medlyn, B. E., Zaehle, S., Walker, A. P., Dietze, M. C., Wang, Y.-P., Luo, Y., Jain, A. K., El-Masri, B., Hickler, T., Wårlind, D., Weng, E., Parton, W. J., Thornton, P. E., Wang, S., Prentice, I. C., Asao, S., Smith, B., McCarthy, H. R., Iversen, C. M., Hanson, P. J., Warren, J. M., Oren, R., and Norby, R. J. (2014). Where does the carbon go? A model–data intercomparison of vegetation carbon allocation and turnover processes at two temperate forest free-air CO2 enrichment sites. *New Phytologist*, 203(3):883–899.

Forkel, M., Carvalhais, N., Rödenbeck, C., Keeling, R., Heimann, M., Thonicke, K., Zaehle, S., and Reichstein, M. (2016). Enhanced seasonal CO2 exchange caused by amplified plant productivity in northern ecosystems. *Science*, 351(6274):696–699.

Friedlingstein, P., Meinshausen, M., Arora, V. K., Jones, C. D., Anav, A., Liddicoat, S. K., and Knutti, R. (2013). Uncertainties in CMIP5 Climate Projections due to Carbon Cycle Feedbacks. *Journal of Climate*, 27:511–526.

Graven, H. D., Keeling, R. F., Piper, S. C., Patra, P. K., Stephens, B. B., Wofsy, S. C., Welp, L. R., Sweeney, C., Tans, P. P., Kelley, J. J., Daube, B. C., Kort, E. A., Santoni, G. W., and Bent, J. D. (2013). Enhanced Seasonal Exchange of CO2 by Northern Ecosystems Since 1960. *Science*, 341(6150):1085–1089.

Keeling, C. D., Chin, J. F. S., and Whorf, T. P. (1996). Increased activity of northern vegetation inferred from atmospheric CO2 measurements. *Nature; London*, 382(6587):146–149.

Kolby Smith, W., Reed, S. C., Cleveland, C. C., Ballantyne, A. P., Anderegg, W. R. L., Wieder, W. R., Liu, Y. Y., and Running, S. W. (2016). Large divergence of satellite and Earth system model estimates of global terrestrial CO2 fertilization. *Nature Climate Change*, 6(3):306–310.

Körner, C., Asshoff, R., Bignucolo, O., Hättenschwiler, S., Keel, S. G., Peláez-Riedl, S., Pepin, S., Siegwolf, R. T. W., and Zotz, G. (2005). Carbon Flux and Growth in Mature Deciduous Forest Trees Exposed to Elevated CO2. *Science*, 309(5739):1360–1362.

Mahowald, N., Lo, F., Zheng, Y., Harrison, L., Funk, C., Lombardozzi, D., and Goodale, C. (2016). Projections of leaf area index in earth system models. *Earth Syst. Dynam.*, 7:211–229.

Mao, J., Ribes, A., Yan, B., Shi, X., Thornton, P. E., Séférian, R., Ciais, P., Myneni, R. B., Douville, H., Piao, S., Zhu, Z., Dickinson, R. E., Dai, Y., Ricciuto, D. M., Jin, M., Hoffman, F. M., Wang, B., Huang, M., and Lian, X. (2016). Human-induced greening of the northern extratropical land surface. *Nature Climate Change*, 6(10):959–963.

Myneni, R., Keeling, C. D., Tucker, C. J., Asrar, G., and Nemani, R. R. (1997). Increased plant growth in the northern high latitudes from 1981 to 1991. *Nature*, 386:698–702.

Mystakidis, S., Davin, E. L., Gruber, N., and Seneviratne, S. I. (2016). Constraining future terrestrial carbon cycle projections using observation-based water and carbon flux estimates. *Global Change Biology*, 22(6):2198–2215.

Norby, R. J., DeLucia, E. H., Gielen, B., Calfapietra, C., Giardina, C. P., King, J. S., Ledford, J., McCarthy, H. R., Moore, D. J. P., Ceulemans, R., Angelis, P. D., Finzi, A. C., Karnosky, D. F., Kubiske, M. E., Lukac, M., Pregitzer, K. S., Scarascia-Mugnozza, G. E., Schlesinger, W. H., and Oren, R. (2005). Forest response to elevated CO2 is conserved across a broad range of productivity. *Proceedings of the National Academy of Sciences*, 102(50):18052–18056.

Peñuelas, J., Ciais, P., Canadell, J. G., Janssens, I. A., Fernández-Martínez, M., Carnicer, J., Obersteiner, M., Piao, S., Vautard, R., and Sardans, J. (2017). Shifting from a fertilization-dominated to a warming-dominated period. *Nature Ecology & Evolution*, 1(10):1438.

Quéré, C. L., Andrew, R. M., Friedlingstein, P., Sitch, S., Pongratz, J., Manning, A. C., Korsbakken, J. I., Peters, G. P., Canadell, J. G., Jackson, R. B., Boden, T. A., Tans, P. P., Andrews, O. D., Arora, V. K., Bakker, D. C. E., Barbero, L., Becker, M., Betts, R. A., Bopp, L., Chevallier, F., Chini, L. P., Ciais, P., Cosca, C. E., Cross, J., Currie, K., Gasser, T., Harris, I., Hauck, J., Haverd, V., Houghton, R. A., Hunt, C. W., Hurtt, G., Ilyina, T., Jain, A. K., Kato, E., Kautz, M., Keeling, R. F., Klein Goldewijk, K., Körtzinger, A., Landschützer, P., Lefèvre, N., Lenton, A., Lienert, S., Lima, I., Lombardozzi, D., Metzl, N., Millero, F., Monteiro, P. M. S., Munro, D. R., Nabel, J. E. M. S., Nakaoka, S.-i., Nojiri, Y., Padin, X. A., Peregon, A., Pfeil, B., Pierrot, D., Poulter, B., Rehder, G., Reimer, J., Rödenbeck, C., Schwinger, J., Séférian, R., Skjelvan, I., Stocker, B. D., Tian, H., Tilbrook, B., Tubiello, F. N., Laan-Luijkx, I. T. v. d., Werf, G. R. v. d., Heuven, S. v., Viovy, N., Vuichard, N., Walker, A. P., Watson, A. J., Wiltshire, A. J., Zaehle, S., and Zhu, D. (2018). Global Carbon Budget 2017. *Earth System Science Data*, 10(1):405–448.

Schimel, D., Stephens, B. B., and Fisher, J. B. (2015). Effect of increasing CO2 on the terrestrial carbon cycle. *Proceedings of the National Academy of Sciences*, 112(2):436–441.

Terrer, C., Vicca, S., Hungate, B. A., Phillips, R. P., and Prentice, I. C. (2016). Mycorrhizal association as a primary control of the CO2 fertilization effect. *Science*, 353(6294):72–74.

Wenzel, S., Cox, P. M., Eyring, V., and Friedlingstein, P. (2014). Emergent constraints on climate-carbon cycle feedbacks in the CMIP5 Earth system models. *Journal of Geophysical Research: Biogeosciences*, 119(5):794–807.

Wenzel, S., Cox, P. M., Eyring, V., and Friedlingstein, P. (2016). Projected land photosynthesis constrained by changes in the seasonal cycle of atmospheric CO2. *Nature*, 538(7626):499–501.

Winkler, A. J., Myneni, R. B., Alexandrov, G. A., and Brovkin, V. (2019). Earth system models underestimate carbon fixation by plants in the high latitudes. *Nature Communications*, 10(1):885.

Zhu, Z., Bi, J., Pan, Y., Ganguly, S., Anav, A., Xu, L., Samanta, A., Piao, S., Nemani, R. R., and Myneni, R. B. (2013). Global Data Sets of Vegetation Leaf Area Index (LAI)3g and Fraction of Photosynthetically Active Radiation (FPAR)3g Derived from Global Inventory Modeling and Mapping Studies (GIMMS) Normalized Difference Vegetation Index (NDVI3g) for the Period 1981 to 2011. *Remote Sensing*, 5(2):927–948.

Zhu, Z., Piao, S., Myneni, R. B., Huang, M., Zeng, Z., Canadell, J. G., Ciais, P., Sitch, S., Friedlingstein, P., Arneth, A., Cao, C., Cheng, L., Kato, E., Koven, C., Li, Y., Lian, X., Liu, Y., Liu, R., Mao, J., Pan, Y., Peng, S., Peñuelas, J., Poulter, B., Pugh, T. A. M., Stocker, B. D., Viovy, N., Wang, X., Wang, Y., Xiao, Z., Yang, H., Zaehle, S., and Zeng, N. (2016). Greening of the Earth and its drivers. *Nature Climate Change*, 6(8):791–795.

---

## Referee Comment (RC2) · Anonymous Referee #2 · 18 Mar 2019

**Review of Winkler et al.:**
Limitations of emergent constraints on multi-model projections: case study of constraining vegetation productivity with observed greening sensitivity

The authors explore the robustness of the "emerging constraint" (EC) method by using vegetation changes in the Northern High Latitudes as a case study. As the authors discuss, the EC method has gained increasing popularity and is being applied to a wide range of climate change studies, including reducing uncertainty in the carbon cycle. Overall the paper is well-written and easy to follow. The authors identify and analyse a number of caveats that may influence results from the EC method that are likely relevant to the wider community.

My main criticism is the use of LAI to predict GPP changes and stating that these two variables possess a strong causal relationship (indeed the authors state that the predictor and predictand should be causally related). Yes LAI and GPP are likely related, but there are many assumptions in models regarding how much GPP becomes NPP (i.e. how much GPP is respired) and how this carbon is then allocated into leaves, as opposed to other plant tissues. Furthermore, in the ESMs, allocation and respiration etc. can change with increased $CO_2$ forcing, influencing the GPP-LAI relationship over time. The authors should at the very least discuss the caveats of this approach and how this might affect their conclusions. I would also like the authors to consider in more detail what aspects of their findings might be specific to their case study (for example the idealised experiments where the effects of radiation and fertilisation effects are rather straightforward and increase GPP).

Specific comments:

P5 L21: Should this say 0.005 deg (500m) instead of 0.05 deg (5km)?

P5 L32: why averaged and not taking the max (to further reduce cloud contamination)?

P6 L26-28: Not very clear

P7 L7: Can you provide a few more details for "w" and how it was derived? Is it the time series for PC1?

P8 L15: Where do the vegetation classes come from?

P8 L17: I'm a little confused here. You talk about NHL but then go on to describe differences in tropical forests etc.

P11 L18: I admit I had to google the meaning of "Gedankenexperiment", perhaps a more common term is available?

P12 L9: do you show this anywhere?

P13 L14: Do these models simulate species composition changes?

P16 L1 onwards: these aren't really results presented in this study

Figure 2: I don't quite follow why only NHL was analysed when boreal, temperate forests and grasslands all show good agreement between AVHRR and MODIS (if this was the premise of the authors' choice)?

---

## Author Comment (AC2) · 19 Mar 2019

**Authors' Response to Referee 2 (ESDD esd-2018-71)**

March 19, 2019

*The authors explore the robustness of the "emerging constraint" (EC) method by using vegetation changes in the Northern High Latitudes as a case study. As the authors discuss, the EC method has gained increasing popularity and is being applied to a wide range of climate change studies, including reducing uncertainty in the carbon cycle. Overall the paper is well-written and easy to follow. The authors identify and analyse a number of caveats that may influence results from the EC method that are likely relevant to the wider community.*

We thank the reviewer for her/his constructive comments and for acknowledging the relevance of our study to a wider scientific community. All revisions done in response to the reviewer's comments has resulted in an improved manuscript.

**1 General Comments**

1.1 *My main criticism is the use of LAI to predict GPP changes and stating that these two variables possess a strong causal relationship (indeed the authors state that the predictor and predictand should be causally related). Yes, LAI and GPP are likely related, but there are many assumptions in models regarding how much GPP becomes NPP (i.e. how much GPP is respired) and how this carbon is then allocated into leaves, as opposed to other plant tissues.*

We agree with the referee, that the current manuscript lacks an in-depth discussion on the causal link between predictor and predictand. However, this aspect is discussed in more detail in the recently published companion paper by Winkler et al. (2019) and illustrated in Supplementary Figure 1 - *Schematic of the Emergent Constraint concept* (see Fig. R2-1). In our responses to Referee 1, we present a comprehensive analysis of the causal relationship of LAI and GPP in observations on the basis of upscaled eddy covariance flux measurements of GPP (FluxNet and FLUXCOM) and satellite observations of LAI (AVHRR and MODIS). For more details, please see Fig. R1-1 and comment 1.7 in our responses to Referee 1. In the model world, as the referee correctly states, there are many, possibly diverging, assumptions on carbon allocation to various plant organs. But overall, the CMIP5 model ensemble agrees on a tight relationship between concurrent changes in LAI and annual mean GPP for the historical period (1860 to 2005) in the NHL (60° N - 90° N; see Fig. R2-2). This strong link between the predictand GPP and the predictor LAI in NHL, as shown for observations and models alike, is the baseline for the EC study in hand. We discuss this aspect in more detail in the revised manuscript.

1.2 *Furthermore, in the ESMs, allocation and respiration etc. can change with increased $CO_2$ forcing, influencing the GPP-LAI relationship over time. The authors should at the very least discuss the caveats of this approach and how this might affect their conclusions.*

Yes, the evolution of the predictand-predictor relationship in the course of the forcing is an essential aspect in the EC method. We already address this issue in the manuscript and is at the very core of the *Gedankenexperiment* discussed in Sect. 3.4 (P11, L2- P12, L33) and Fig. 5, 6, A1, and A2. The GPP-LAI relationship likely changes with increasing $CO_2$ as predicted by CMIP5 models (saturation of GPP to LAI allocation above 2×$CO_2$, Fig. 5). In the *Gedankenexperiment*, we conceive four possible scenarios of how the system might behave with increasing forcing. We show that changes in predictor and predictand relate linearly within the model ensemble (the basis for an EC) given the models agree on the occurrence and "timing" of saturation. At very high $CO_2$

[Figure]

Figure R2- 1: Schematic of the Emergent Constraint concept. The radiative and physiological effects of increasing atmospheric $CO_2$ concentration, in the range 280 to 560 ppm, are thought to increase GPP. This is indirectly observed as changes in LAI or the amplitude of the seasonal cycle of atmospheric $CO_2$. The sensitivity of changes in observables to historical increase in $CO_2$ concentration (e.g., 280 to 400 ppm) can be thought of as an Emergent Constraint on model-projected changes in carbon cycle quantities (e.g., $\Delta$GPP for $CO_2$ change from 280 to 560 ppm), if the inter-model variation of projections is linear, or nearly so, with respect to modelled historical sensitivities. GPP enhancement from the positive feedback effect (blue arrow) is thought to be small relative to the physiological and radiative effects (Keenan et al., 2016). Supplementary Figure 1 in Winkler et al. (2019).

concentrations (above 3x$CO_2$), this is not the case anymore in the CMIP5 ensemble resulting in a weakening of the relationship between GPP and LAI.

1.3 *I would also like the authors to consider in more detail what aspects of their findings might be specific to their case study (for example the idealised experiments where the effects of radiation and fertilisation effects are rather straightforward and increase GPP)*

Each Emergent Constraint is somewhat unique in its mechanistic relationship under a strengthening forcing. However, in theory, the results presented in this study are qualitatively transmissive to other sets of predictors and predictands. Of course, the aspect related to idealized setups of disentangling radiative and fertilizing effects of $CO_2$ is rather specific for carbon cycle research. Other aspects, such as the the influence of the observational estimate (dependence on observational source), predictor comparability between models and observations (especially within the temporal domain), or uncertainty based on spatial aggregation of gridded data are more general. We included a short paragraph discussing general and more specific findings in the revised manuscript.

**2 Specific comments:**

2.1 *P5, L21: Should this say 0.005 deg (500m) instead of 0.05 deg (5km)?*

MODIS LAI products (Collection 6, Aqua and Terra) are provided as 8-day composites with a 500m sinusoidal projection covering the whole globe. To minimize contamination from clouds, aerosols, snow and shadow, careful quality assurance and filtering techniques are applied to obtain highest quality MODIS LAI observations. The 16-day composite LAI dataset is then derived by taking the

[Figure]

Figure R2- 2: Linear relationship between concurrent changes in $LAI_{max}$ and annual mean GPP. Comparison of changes in $LAI_{max}$ and annual mean GPP for the historical period (1860 to 2005) for the NHL (60° N - 90° N) in the CMIP5 ensemble. The colored dots show values for 30 year chunks of the total time series (error bars denote one standard deviation). The colored lines represent the best linear fit for each model, while the black line indicates the best linear fit for all models. The 68% confidence interval estimated by bootstrapping is shown by the grey shading. Supplementary Figure 2 in Winkler et al. (2019).

mean of all valid LAI estimates from the 8-day composites. The final dataset is provided at a spatially aggregated 0.05° climate-modelling grid (CMG; Chen et al., 2019).

**2.2 *P5, L32: why averaged and not taking the max (to further reduce cloud contamination)?**

Comparability between models and observations are key in the EC method. CMIP5 models provide LAI as monthly means. Averaging the 16-day composites for each month is the closest we get to a monthly mean estimate from observations.

**2.3 *P6, L26-28: Not very clear**

Yes, we agree. The description of the idealized CMIP5 simulations is somewhat confusing. We rewrote this section in the revised manuscript.

**2.4 *P7, L7: Can you provide a few more details for "$\omega$" and how it was derived? Is it the time series for PC1?**

We agree, that a more detailed description of the derivation of $\omega$ needs to be provided. We perform a PCA of the time-series of $CO_2$ and GDD0 on large-scale aggregated values as well as on pixel level to investigate on spatial variations. We only retain the first principal component (denoted $\omega$), which explains a large fraction of the variance, ranging approximately from 70% ot 90% in models and observations (for more details see Table R1-1 in our responses to Referee 1). Figure R2-3 depicts the temporal development of $CO_2$ and GDD0 as well as the principal component $\omega$ for observations. Please see also our responses to comment 1.2 and 1.5 of Referee 1. In the

revised manuscript, we make abundantly clear how $\omega$ is derived and describe its characteristics for models and observations. Figure R2-3, with some modifications, has been included in the appendix of the revised manuscript.

[Figure]

Figure R2- 3: Standardized temporal anomalies of annual averaged atmospheric $CO_2$ concentration, area-weighted averaged GDD0 for NHL, and their leading principal component $\omega$ in observations.

**2.5 *P8, L15: Where do the vegetation classes come from?**

We provide the reference (Olson et al., 2001) in Sect. 2.3 **Estimation of greening sensitivities** (P6, L32) and in the caption of Figure 2.

**2.6 *P8, L17: I'm a little confused here. You talk about NHL but then go on to describe differences in tropical forests etc.**

In Sect. 3.1 **Uncertainty in Observed Sensitivity Due to Data Source** we present sensitivities in a global comparison for different climatic regimes, vegetation types, and latitudinal bands (e.g. comparing tropical, mid-latitude, and high-latitude sensitivities; Figure 2). Then, we show that LAI is only a meaningful predictor for changes in GPP in the northern high latitudes, which constitutes the focus of the study thereafter. We rewrote this section and provide better explanation for our approach.

**2.7 *P11, L18: I admit I had to google the meaning of "Gedankenexperiment", perhaps a more common term is available?**

Yes, in the revised manuscript we now use the term "thought experiment".

**2.8 *P12, L9: do you show this anywhere?**

Figure 4 and Table 2 in the manuscript illustrate that the CMIP5 models (3 models are shown) reveal saturation of the relationship between $\Delta LAI_{max}$ and $\Delta GPP$ with increasing $CO_2$ forcing. The slopes in Figure 4 (detailed estimates in Table 2) reveal that the strength and 'timing' of saturation (i.e. at what level of $CO_2$ concentration) differs among the models. In the revised manuscript, we implemented a more accurate description and references to tables and figures. Also, we generated an additional figure (shown in the appendix of the revised manuscript) which displays the results of the other 4 models analogous to Figure 4.

**2.9 *P13, L14: Do these models simulate species composition changes?**

Yes, most of the models include dynamic vegetation. In the revised manuscript, we include a short description of the representation of dynamic vegetation in CMIP5 models. In general, the historical and idealized model setups of the CMIP5 land components are comprehensively explained in several studies, such as Wenzel et al. (2014); Mahowald et al. (2016); Arora et al. (2013); Winkler et al. (2019). This is why we refrain from providing a detailed overview of the CMIP5 models in this study.

**2.10 *P16, L1 onward: These aren't really results presented in this study**

Yes, we agree, these are rather findings presented in the companion paper by Winkler et al. (2019). We rewrote this paragraph to sharpen the focus on the results discussed in this article.

**2.11 *Figure 2: I don't quite follow why only NHL was analysed when boreal, temperate forests and grasslands all show good agreement between AVHRR and MODIS (if this was the premise of the authors' choice)?**

Ecosystems in NHL are barely influenced by human land use. Thus, the changes of vegetation greenness are natural responses to the forcing rather than agricultural artifacts. At high LAI regions, GPP might also increase due to $CO_2$ fertilization without an enhancement of LAI. In rural areas, the observed greening is mainly caused by direct drivers such irrigation, application of fertilizers, and double cropping as shown recently by Chen et al. (2019). Overall, we focus on the NHL, because there we obtain a clear LAI signal, i.e. a signal hardly being distorted by direct human interference.

**References**

Arora, V. K., Boer, G. J., Friedlingstein, P., Eby, M., Jones, C. D., Christian, J. R., Bonan, G., Bopp, L., Brovkin, V., Cadule, P., Hajima, T., Ilyina, T., Lindsay, K., Tjiputra, J. F., and Wu, T. (2013). Carbon–Concentration and Carbon–Climate Feedbacks in CMIP5 Earth System Models. *Journal of Climate*, 26:5289–5314.

Chen, C., Park, T., Wang, X., Piao, S., Xu, B., Chaturvedi, R. K., Fuchs, R., Brovkin, V., Ciais, P., Fensholt, R., Tømmervik, H., Bala, G., Zhu, Z., Nemani, R. R., and Myneni, R. B. (2019). China and India lead in greening of the world through land-use management. *Nature Sustainability*, 2(2):122.

Keenan, T. F., Prentice, I. C., Canadell, J. G., Williams, C. A., Wang, H., Raupach, M., and Collatz, G. J. (2016). Recent pause in the growth rate of atmospheric CO2 due to enhanced terrestrial carbon uptake. *Nature Communications*, 7:13428.

Mahowald, N., Lo, F., Zheng, Y., Harrison, L., Funk, C., Lombardozzi, D., and Goodale, C. (2016). Projections of leaf area index in earth system models. *Earth Syst. Dynam.*, 7:211–229.

Olson, D. M., Dinerstein, E., Wikramanayake, E. D., Burgess, N. D., Powell, G. V. N., Underwood, E. C., D'amico, J. A., Itoua, I., Strand, H. E., Morrison, J. C., Loucks, C. J., Allnutt, T. F., Ricketts, T. H., Kura, Y., Lamoreux, J. F., Wettengel, W. W., Hedao, P., and Kassem, K. R. (2001). Terrestrial Ecoregions of the World: A New Map of Life on Earth. *BioScience*, 51(11):933–938.

Wenzel, S., Cox, P. M., Eyring, V., and Friedlingstein, P. (2014). Emergent constraints on climate-carbon cycle feedbacks in the CMIP5 Earth system models. *Journal of Geophysical Research: Biogeosciences*, 119(5):794–807.

Winkler, A. J., Myneni, R. B., Alexandrov, G. A., and Brovkin, V. (2019). Earth system models underestimate carbon fixation by plants in the high latitudes. *Nature Communications*, 10(1):885.

---

## Referee Comment (RC3) · Anonymous Referee #3 · 26 Mar 2019

Please accept my apologies for this very slow response to reviewing: "Limitations of Emergent Constraints on Multi-Model Projections: Case Study of Constraining Vegetation Productivity With Observed Greening Sensitivity"

Emergent Constraints (ECs) have become a very popular mechanism to collapse inter-GCM differences, and in order to make more refined future projections. It is therefore highly relevant to verify how robust the methodology is, and/or find counter-examples which illustrate potential issues with the technique.

[Figure]

This is a slightly superficial review, but what I would encourage the authors to do is to focus more tightly on the issue of potential problems with ECs – maybe at the expense of some of the other text describing so fully the particulars of vegetation greening.

While there are some concerns surrounding the EC approach, some of the criticisms levelled by the authors are only valid if the approach is applied carelessly. So I am not convinced these are limitations, and instead, a better title might be "Careful Application needed by ECs…..".

The Abstract raises two concerns.

(*) "The method critically depends on first an accurate estimation of the predictor from observations and models". This is true, but this is not particular to ECs any more than it is for any other environmental science modelling exercise. It is always essential to ensure that measurements align tightly with models to – for instance – allow model calibration. For example, the need for "like-for-like" comparison is routinely addressed when utilising Earth Observing data to constrain terrestrial ecosystem models.

(*) "Second, depends on a robust relationship between inter-model variations in the predictor-predictand space". This is really what lies at the heart of emergent constraints, which by definition is the search for emerging regressions across "X" and "Y"-axis space. However, if no relationship is present, then clearly the method would not be used. An interesting question to ask, however, is if intuitively a relationship is expected, but is not seen inter-GCM, then what does this imply?

The Conclusions are much more nicely set out, and I think clearer to understand. However, to just run through the points raised:

(*) The paragraph starting "The importance of how the observational predictor….." again raises the need for all EC modellers to ensure a direct 1-1 mapping between modelled "X"-axis quantity and measurements. The next paragraph correctly identifies the importance of accurate spatial aggregation, when the GCMs themselves are predicting bulk quantities, defined as only valid over large regions (e.g. mean "greening" levels").

(*) "A large source of uncertainty is associated with temporal variability". The EC method does account for uncertainty in the measured "X"-quantity, which is why the standard diagrams place bounds on that – in addition to uncertainty associated with the model-based regression. If only one measurement is available, based on averaging over multiple years, then standard statistical techniques can be used to build error bounds. These can include, for instance, sampling only subsets of the years. Methods like this can also be applied where there is a mismatch in window length, to ensure larger uncertainty bounds where the measured quantity is over a short period.

(*) The conclusion hints at the issue of the importance of both identical "X"-axis temporal length (both model and measurement), and additionally the need for identical time-periods. In its most extreme for instance, it would not be appropriate to take 30-year segments of GCM period 1850-1889, comparing to 1990-2019 measurements. This is because an EC can change in time. Such variation is sometimes used to question ECs, but as long as the "X" model and "X" data are for the same period, then the method remains valid.

(*) Indeed here, the authors acknowledge dGPP v dLAI_max relationships do change for increasing CO2 levels. These changes are not a failure of the EC method, simply that (i) timescales need to line up correctly for present day (data versus models), and (ii) users need to be aware of what CO2 level is being considered for the "Y"-axis.

(*) I think the authors might miss a trick here, and especially for vegetation analysis. Where the EC approach is at risk of failure is if all GCMs currently miss an important process, and that will only become critical into the future. One prominent example is where, until recently at least, very few GCMs describe possible future down-regulation of fertilisation through geochemical cycles such as that of Nitrogen.

I certainly do not want this review to appear defensive of ECs, and this is indeed a very

interesting and thought-provoking manuscript. There are definitely things that require investigation associated with the technique. It is just that most of the points raised do not invalidate the EC approach – the examples are much more a case of "please use ECs carefully"?

Sorry, this is a short review, but if another version if generated then I would be happy to see the paper again.

---

## Author Comment (AC3) · 31 Mar 2019

**Authors' Response to Referee 3 (ESDD esd-2018-71)**

March 31, 2019

*Please accept my apologies for this very slow response to reviewing: "Limitations of Emergent Constraints on Multi-Model Projections: Case Study of Constraining Vegetation Productivity With Observed Greening Sensitivity" Emergent Constraints (ECs) have become a very popular mechanism to collapse inter-GCM differences, and in order to make more refined future projections. It is therefore highly relevant to verify how robust the methodology is, and/or find counter-examples which illustrate potential issues with the technique.*

We thank the reviewer for her/his interesting comments on the fundamentals of the EC method and for sharing the opinion, that scrutinizing the EC methodology towards robustness is highly relevant. Apparently, the reviewer also got the notion that the purpose of this study is to question the general validity of the EC approach (see comment 1.7). This is not case. In fact, we illustrate the range of applicability of ECs and elaborate on caveats and potential pitfalls. We revised the manuscript to avoid such misunderstandings.

**1 General Comments**

*1.1 This is a slightly superficial review, but what I would encourage the authors to do is to focus more tightly on the issue of potential problems with ECs – maybe at the expense of some of the other text describing so fully the particulars of vegetation greening.*

Yes, we agree with the reviewer. In the revised manuscript we bring the general applicability of EC more into focus. We encapsulated the particulars of vegetation greening, but keep a certain detail, so that the reader can easily follow the narrative of the article.

*1.2 While there are some concerns surrounding the EC approach, some of the criticisms levelled by the authors are only valid if the approach is applied carelessly. So I am not convinced these are limitations, and instead, a better title might be "Careful Application needed by ECs. …".*

As stated in the response to the reviewer's summary comment, the intention of this study is not to question the general EC approach, but rather challenge its robustness and illustrate caveats. But we agree with the reviewer, the manuscript is mostly dealing with the applicability of ECs and that inaccuracies in the methodology can crucially influence conclusions drawn from the constrained estimate. We adjusted the title in the revised manuscript.

*1.3 The Abstract raises two concerns. "The method critically depends on first an accurate estimation of the predictor from observations and models". This is true, but this is not particular to ECs any more than it is for any other environmental science modelling exercise. It is always essential to ensure that measurements align tightly with models to – for instance – allow model calibration. For example, the need for "like-for-like" comparison is routinely addressed when utilising Earth Observing data to constrain terrestrial ecosystem models.*

Yes, observational uncertainty is an important issue in many statistical methods, and in calibrating or benchmarking environmental numerical models. However, we argue, that the EC method is particularly sensitive to observational uncertainty (P15, L5-6), because the single observational estimate essentially determines the EC. On the contrary, the emergent linear relationship is established based on a collection of multi-model estimates, where each model gets 'one vote' (however, some models might be more influential than others; Bracegirdle and Stephenson, 2012). Thus, the

observational uncertainty has a much larger bearing on the EC than the uncertainty of each individual model. To overcome this source of uncertainty, various meaningful observations, if applicable, should be taken into consideration.

1.4 *"Second, depends on a robust relationship between inter-model variations in the predictor-predictand space". This is really what lies at the heart of emergent constraints, which by definition is the search for emerging regressions across "X" and "Y"-axis space. However, if no relationship is present, then clearly the method would not be used. An interesting question to ask, however, is if intuitively a relationship is expected, but is not seen inter-GCM, then what does this imply?*

The reviewer raises an interesting issue here. In the 'search' for an emergent linear relationship, the researcher might stumble upon one or two predictor-predictand combinations that do not show a tight connection, albeit it was expected based on the current process understanding. This implies that (some) models miss or misrepresent the process of interest, assuming that indeed a meaningful predictor was chosen. For the sake of model development and advancement, such non-existent, yet expected, Emergent Constraints should also be reported and scrutinized to find out why the individual models deviate. We implement this aspect in the discussion section in the revised manuscript.

1.5 *The Conclusions are much more nicely set out, and I think clearer to understand. However, to just run through the points raised: (\*) The paragraph starting "The importance of how the observational predictor....." again raises the need for all EC modellers to ensure a direct 1-1 mapping between modelled "X"-axis quantity and measurements. The next paragraph correctly identifies the importance of accurate spatial aggregation, when the GCMs themselves are predicting bulk quantities, defined as only valid over large regions (e.g. mean "greening" levels). A large source of uncertainty is associated with temporal variability". The EC method does account for uncertainty in the measured "X"-quantity, which is why the standard diagrams place bounds on that – in addition to uncertainty associated with the model-based regression. If only one measurement is available, based on averaging over multiple years, then standard statistical techniques can be used to build error bounds. These can include, for instance, sampling only subsets of the years. Methods like this can also be applied where there is a mismatch in window length, to ensure larger uncertainty bounds where the measured quantity is over a short period.*

Yes, the EC method accounts for temporal fluctuations, for models and observations alike. However, the signal-to-noise ratio changes with increasing forcing, i.e. the predictor is strongly influenced by temporal fluctuations at a low $CO_2$ forcing (Figure 3). If these fluctuations are taken into account for the modeled predictors, but not for the observational estimate (due to the lack of certain observations at low $CO_2$ forcing), predictor comparability is not given, which results in a questionable constrained estimate.

1.6 *The conclusion hints at the issue of the importance of both identical "X"-axis temporal length (both model and measurement), and additionally the need for identical time-periods. In its most extreme for instance, it would not be appropriate to take 30-year segments of GCM period 1850-1889, comparing to 1990-2019 measurements. This is because an EC can change in time. Such variation is sometimes used to question ECs, but as long as the "X" model and "X" data are for the same period, then the method remains valid. Indeed here, the authors acknowledge dGPP v dLAI_max relationships do change for increasing $CO_2$ levels. These changes are not a failure of the EC method, simply that (i) timescales need to line up correctly for present day (data versus models), and (ii) users need to be aware of what $CO_2$ level is being considered for the "Y"-axis. I think the authors might miss a trick here, and especially for vegetation analysis. Where the EC approach is at risk of failure is if all GCMs currently miss an important process, and that will only become critical into the future. One prominent example is where, until recently at least, very few GCMs describe possible future down-regulation of fertilisation through geochemical cycles such as that of Nitrogen.*

Yes, the EC approach is prone to fail if the majority of models miss an essential process, that might critically influence the development of predictor and predictand with increasing forcing. We discuss this aspect in Section 3.6 **Uncertainties in the multi-model ensemble** in the manuscript (P14, L3-4). Also, we addressed the prominent case of potential nitrogen limitation in a high $CO_2$ world in the companion paper by Winkler et al. (2019); discussed in length in the *Peer Review*

*File* https://doi.org/10.1038/s41467-019-08633-z. In theory, the EC relationship should approximately stay constant if more models included a well-calibrated interactive nitrogen cycle. For example, if the HadGEM2-ES model implemented a reasonable nitrogen limitation, estimates of historical greening sensitvitiy (predictor) as well as of future GPP increase (predictand) would be lower, thus, HadGEM2-ES would move down the EC slope approaching the constraint estimate (see Fig.2c in Winkler et al., 2019).

*1.7 I certainly do not want this review to appear defensive of ECs, and this is indeed a very interesting and thought-provoking manuscript. There are definitely things that require investigation associated with the technique. It is just that most of the points raised do not invalidate the EC approach – the examples are much more a case of "please use ECs carefully"? Sorry, this is a short review, but if another version if generated then I would be happy to see the paper again.*

Please see our response to the reviewer's summary comment and to comment 1.2. We thank the reviewer for acknowledging again the relevance of this manuscript and pointing out its thought-provoking character. We are also thankful that the reviewer is willing to see the revised manuscript again.

**References**

Bracegirdle, T. J. and Stephenson, D. B. (2012). On the Robustness of Emergent Constraints Used in Multimodel Climate Change Projections of Arctic Warming. *Journal of Climate*, 26:669–678.

Winkler, A. J., Myneni, R. B., Alexandrov, G. A., and Brovkin, V. (2019). Earth system models underestimate carbon fixation by plants in the high latitudes. *Nature Communications*, 10(1):885.

---

## Author Response (AR1)

Max-Planck-Institut für Meteorologie | Bundesstr. 53 | 20146 Hamburg

**Alexander Winkler**

The Land in the Earth System

Max-Planck-Institut für Meteorologie

Bundesstr. 53

20146 Hamburg

Deutschland

Tel.: +49 - (0)40 - 41173 - 542

alexander.winkler@mpimet.mpg.de www.mpimet.mpg.de

Dr. Vivek Arora

Earth System Dynamics

Copernicus Publications

Hamburg, den 06. June 2019

Dear Dr. Arora,

On behalf of my co-authors, I would like to resubmit the revised manuscript "esd-2018-71" titled "Investigating the Applicability of Emergent Constraints" for publication in Earth System Dynamics. All referee comments are addressed comprehensively with additional analyses and revisions to the manuscripts, as described in the attached point-by-point response files.

Please find below the revised manuscript in marked-up version with revisions highlighted in red and blue.

Thank you for your consideration.

Yours sincerely,

Alexander Winkler (for all co-authors)

[revised manuscript text omitted]

---

## Author Response (AR2)

**Alexander Winkler**

The Land in the Earth System
Max-Planck-Institut für Meteorologie
Bundesstr. 53
20146 Hamburg
Deutschland
Tel.: +49 - (0)40 - 41173 - 542

alexander.winkler@mpimet.mpg.de
www.mpimet.mpg.de

Max-Planck-Institut für Meteorologie | Bundesstr. 53 | 20146 Hamburg

Dr. Vivek Arora
Earth System Dynamics
Copernicus Publications

Hamburg, den 11. July 2019

Dear Dr. Arora,

On behalf of my co-authors, I would like to resubmit the revised manuscript "esd-2018-71" titled "Investigating the Applicability of Emergent Constraints" for publication in Earth System Dynamics. We addressed all of your minor corrections. Regarding your comment on Page 9 Line 29: The number of 36% for the non-significant changing part is correct, because 54% show increasing and 10% show decreasing trends.

Please find below the revised manuscript in marked-up version with revisions highlighted in red and blue.

Thank you for your consideration.

Yours sincerely,

Alexander Winkler (for all co-authors)

[revised manuscript text omitted]